# PROGRAM SYNTHESIS BENCHMARK FOR VISUAL PROGRAMMING IN XLOGOONLINE ENVIRONMENT

## ABSTRACT

Large language and multimodal models have shown remarkable success on various benchmarks focused on specific skills such as general-purpose programming, natural language understanding, math word problem-solving, and visual question answering. However, it is unclear how well these models perform on tasks that require a combination of these skills. In this paper, we curate a novel program synthesis benchmark based on the real-world tasks in the XLogoOnline visual programming environment. Each task requires a combination of different skills such as spatial planning, basic programming, and logical reasoning. Our evaluation shows that current state-of-the-art models like GPT-4V and Llama3-70B struggle to solve these tasks, achieving only 20% and 2.35% success rates, respectively. Next, we develop a fine-tuning pipeline to boost the performance of models by leveraging a large-scale synthetic training dataset with over $80,000$ tasks. Moreover, we showcase how emulator-driven feedback can be used to design a curriculum over training data distribution, through which a fine-tuned Llama3-8B drastically outperforms GPT-4V and Llama3-70B models. Finally, we provide an in-depth failure analysis to understand the limitations of different models. We will publicly release the benchmark for future research on program synthesis in visual programming.

## 1 INTRODUCTION

In recent years, large models have shown remarkable performance in various domains, such as general-purpose programming and visual question answering (Bubeck et al., 2023). For instance, in programming, numerous tools and models use large language models (LLMs) for code generation (Chen et al., 2021; GitHub, 2021) and programming feedback generation (Phung et al., 2024; 2023a;b), revolutionizing how programmers write code and how teachers instruct programming (Peng et al., 2023; Denny et al., 2024). Beyond text-based tasks, the focus has expanded to multimodal models that process and generate not only text but also images, achieving significant success in domains such as visual question answering (Radford et al., 2021) and text-to-image generation (Ramesh et al., 2021).

Despite these successes, the performance of large models on tasks that require a combination of skills remains unclear. Real-world tasks often demand a blend of skills. For example, a typical task like "navigating to the kitchen to fetch ten apples" involves spatial reasoning to understand the environment and plan a path around obstacles, together with basic arithmetic to ensure that exactly ten apples are retrieved. This example illustrates the multifaceted nature of real-world tasks. While various benchmarks focus on specific skills (Chen et al., 2021; Hendrycks et al., 2021c;b; Lin et al., 2022), there is a lack of benchmarks evaluating how large models perform on tasks that require a combination of different skills.

To bridge this gap, we introduce XLOGOMINIPROG, a benchmark for program synthesis in the visual programming domain. Our benchmark is constructed using the Mini-level of the XLogoOnline platform (XLogoOnline, 2024), featuring 85 real-world and $1,000$ synthetic visual programming tasks, each demanding a blend of diverse skills. Figure 1 illustrates examples of these tasks. Each task includes a visual grid with a turtle that needs to be directed to complete a specific goal. For example, in Task 28, the goal is to direct the turtle to collect all red shapes without stepping on the color green, requiring logical reasoning, spatial reasoning, planning, and basic programming skills. Task 38 requires additional math word problem-solving to collect 10 strawberries. These

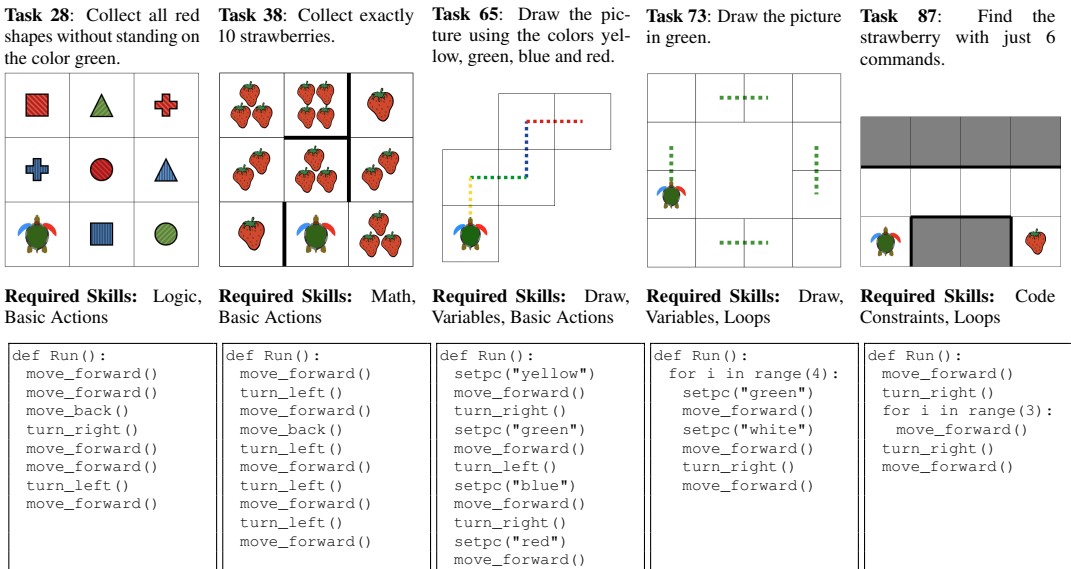

**Task 28**: Collect all red shapes without standing on the color green.

**Required Skills**: Logic, Basic Actions

```
def Run():
  move_forward()
  move_forward()
  move_back()
  turn_right()
  move_forward()
  move_forward()
  turn_left()
  move_forward()
```

**Task 38**: Collect exactly 10 strawberries.

**Required Skills**: Math, Basic Actions

```
def Run():
  move_forward()
  turn_left()
  move_forward()
  move_back()
  turn_left()
  move_forward()
  turn_left()
  move_forward()
  turn_left()
  move_forward()
```

**Task 65**: Draw the picture using the colors yellow, green, blue and red.

**Required Skills**: Draw, Variables, Basic Actions

```
def Run():
  setpc("yellow")
  move_forward()
  turn_right()
  setpc("green")
  move_forward()
  turn_left()
  setpc("blue")
  move_forward()
  turn_right()
  setpc("red")
  move_forward()
```

**Task 73**: Draw the picture in green.

**Required Skills**: Draw, Variables, Loops

```
def Run():
  for i in range(4):
    setpc("green")
    move_forward()
    setpc("white")
    move_forward()
    turn_right()
    move_forward()
```

**Task 87**: Find the strawberry with just 6 commands.

**Required Skills**: Code Constraints, Loops

```
def Run():
  move_forward()
  turn_right()
  for i in range(3):
    move_forward()
  turn_right()
  move_forward()
```

Figure 1: Examples of real-world tasks, required skills, and solution codes in XLogoOnline-Mini.

tasks provide a testbed for evaluating how large models perform on tasks that require a combination of skills, presenting a unique challenge to current large models.

We evaluate the performance of large models on these tasks and find that GPT-4V (Vision) model (OpenAI, 2023b) achieves a 20% success rate on the real-world tasks, and Llama3-70B model (Meta, 2024) struggles significantly, achieving only a 2.35% success rate. This indicates that current large models are not yet capable of effectively solving visual programming tasks requiring various skills. Figure 2 compares the performance of large models across different skill dimensions on these tasks. To improve performance, we develop a fine-tuning pipeline by leveraging a large-scale synthetic dataset containing over 80,000 visual programming tasks. Our fine-tuned Llama3-8B model outperforms GPT-4V and Llama3-70B, achieving a 54.12% success rate. Moreover, we leverage emulator feedback to design a curriculum over the training data distribution, improving performance by 6.1% over standard supervised fine-tuning.

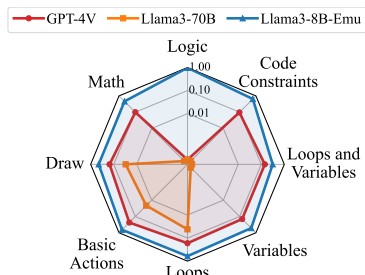

Figure 2: Large models' performance across different skills in real-world tasks (log scale).

Our contributions are as follows: First, we introduce XLOGOMINIPROG, a program synthesis benchmark based on the XLogoOnline platform to evaluate large models in visual programming, which requires a blend of different skills. Second, we develop a fine-tuning pipeline that includes synthetic dataset generation and supervised fine-tuning, along with an emulator-driven fine-tuning technique that improves standard supervised fine-tuning performance by 6.1%. Third, we conduct extensive experiments to benchmark the performance of different models, providing an in-depth failure analysis and a detailed analysis of their expertise across multiple skill dimensions.

## 2 RELATED WORK

**Program synthesis benchmarks for large models.** Program synthesis aims to automatically generate programs from specifications. Recently, numerous recent works have focused on training large models specifically for program synthesis (Chen et al., 2021; Rozière et al., 2023; Fried et al., 2023; Nijkamp et al., 2023). To evaluate these large models, many program synthesis benchmarks have been developed, such as HumanEval (Chen et al., 2021), MBPP (Austin et al., 2021), and APPS (Hendrycks et al., 2021a). However, these benchmarks focus on generating code from natural language or docstrings for general programming languages such as Python (Chen et al., 2021; Austin

et al., 2021; Hendrycks et al., 2021a). Our benchmark focuses on program synthesis in the visual programming domain. While our benchmark covers basic programming like loops and variables, it requires models to combine spatial, logical, and programming skills, posing unique challenges not addressed by these program synthesis benchmarks.

**Large models for visual programming.** Visual programming has been studied in various scenarios, such as task synthesis (Ahmed et al., 2020; Ghosh et al., 2022; Wen et al., 2024; Pădurean et al., 2023), program synthesis (Bunel et al., 2018; Chen et al., 2019b), and student modeling (Nguyen et al., 2024). With the rise of large models, some initial works evaluate ChatGPT (OpenAI, 2023a) and GPT-4 (OpenAI, 2023b) in these scenarios, showing that large models struggle with visual programming tasks (Pădurean et al., 2023; Nguyen et al., 2024; Singla, 2023). In contrast, we provide a comprehensive benchmark for evaluating large models for program synthesis in visual programming, considering a wider range of models and skills.

**Spatial reasoning and planning benchmarks.** Existing benchmarks for spatial reasoning and planning are primarily designed for reinforcement learning agents to solve sequential decision-making tasks (Chevalier-Boisvert et al., 2019; 2023). Additionally, some benchmarks aim to evaluate models in domains where spatial reasoning and planning skills are essential, such as visual navigation and object interaction (Shridhar et al., 2020; Chen et al., 2019a). With the advent of large models, recent works have also begun to evaluate LLMs' capabilities in spatial reasoning and planning (Aghzal et al., 2023; Valmeekam et al., 2023). Our benchmark, however, focuses on the visual programming domain, which requires a broader range of skills beyond spatial reasoning and planning, including logical reasoning, math word problem-solving, and programming skills.

## 3 BACKGROUND AND SYNTHESIS OBJECTIVE

In this section, we provide the background on the XLogoOnline visual programming platform and then introduce the program synthesis objective.

### 3.1 BACKGROUND ON XLOGOONLINE-MINI PROGRAMMING

XLogoOnline (XLogoOnline, 2024) is a visual programming platform based on Logo programming language (Pea, 1987) and is widely used by tens of thousands of students every year (Hromkovic et al., 2017; Staub, 2021). In this work, we focus on the Mini-level (XLogoOnline-Mini). In XLogoOnline-Mini, each task includes a text description of the goal and code constraints, along with a two-dimensional visual grid. The visual grid features a turtle and various elements such as fruits, shapes, colors, lines, walls, and forbidden areas. To solve the task, one needs to write a program to direct the turtle's movement in the visual grid to achieve the specified goal. Figure 1 shows illustrative examples of tasks, the required skills, and solution codes.

**Required skills for XLogoOnline-Mini.** We examine the skills required for solving visual programming tasks in XLogoOnline-Mini. Specifically, the visual programming tasks in our domain cover the following skills: (i) *Logic*: Understand underlying logical relationships specified in the goal; (ii) *Math*: Apply basic arithmetic to solve the task; (iii) *Draw*: Identify patterns and generate the corresponding code; (iv) *Basic actions*: Move and change directions using only basic commands; (v) *Loops*: Utilize loops to repeat commands multiple times; (vi) *Variables*: Utilize variables to set and update colors to draw lines with a specific color; (vii) *Loops and Variables*: Integrate loops with variables to solve a task; (viii) *Code Constraints*: Adhere to specific code constraints such as maximum code length.

### 3.2 PROGRAM SYNTHESIS OBJECTIVE

Next, we formally define task and code specifications, and introduce our synthesis objective.

**Task specifications.** In XLogoOnline-Mini, a task $T := (G, L, W)$ consists of a goal $G$, code constraints $L$, and a visual grid world $W$. The goal $G$ defines the turtle's objective. The code constraints $L$ specify the requirements for a solution code. There are five types of constraints for code: `None` (no restrictions), `AtMost` (maximum number of commands), `Exactly` (exact number of commands), `StartBy` (initial command sequence), and `Hybrid` (combination of constraints). The visual grid world $W$ is a 2-dimensional visual grid featuring a turtle and various elements. We define the grid size as the maximum dimension of the grid. For example, in Figure 1 (Task 87), the goal is "Find

| Task Type | # | Code Constraints | # | Code Concepts | # | Code Length | # | Grid Size | # |
|---|---|---|---|---|---|---|---|---|---|
| Find | 33 | None | 54 | Basic Actions | 47 | Short (1-5) | 41 | $Size \leq 3$ | 59 |
| Draw | 33 | AtMost | 21 | Loops | 24 | Medium (6-10) | 29 | $Size = 4$ | 15 |
| Math | 10 | Exactly | 6 | Variables | 7 | Long (11-17) | 15 | $Size = 5$ | 4 |
| Logic | 9 | StartBy | 4 | Loops and Variables | 7 | | | $Size = 6$ | 4 |
| | | Hybrid | 0 | | | | | $Size \geq 7$ | 3 |
| Total | 85 | Total | 85 | Total | 85 | Total | 85 | Total | 85 |

(a) Task distribution of REAL dataset.

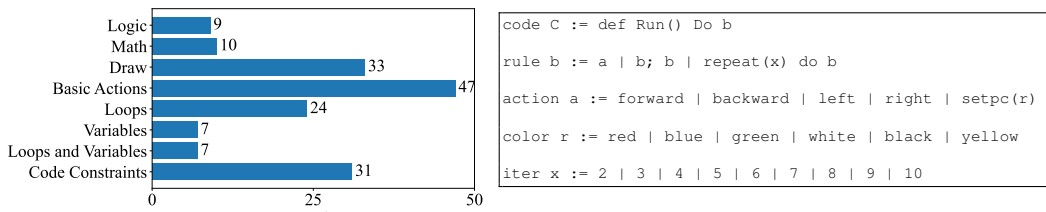

(b) Skill distribution of REAL dataset.          (c) Code DSL.

Figure 3: Statistics of the REAL dataset and the code DSL. (a) shows the task distribution across five dimensions within REAL. (b) illustrates the skill distribution. To describe these skills, we extract various aspects from task type, code constraints, and code concepts as detailed in (a), and consolidated these aspects into broader categories, which we refer to as *skills*. A task may require multiple skills (see Figure 1). (c) shows the code DSL used in the XLogoOnline-Mini domain.

the strawberry", the code constraint is "use just 6 commands" (AtMost), and the visual grid world depicts a $3 \times 4$ grid ($size = 4$) with a turtle, a strawberry, and forbidden areas marked by gray cells.

**Code specifications.** The code space of XLogoOnline-Mini is defined by the domain-specific language (DSL) depicted in Figure 3c. Note that while the DSL defines the formal structure and syntax, we implement it using Python-style code representation to leverage the large models' pre-trained knowledge on Python programming. A *solution code* for a given task is the code that meets the task's code constraints and achieves the specified goal when executed in the visual grid world. In Figure 1, a solution code is provided below each task.

**Program synthesis objective.** Our objective is to develop a synthesizer function, $f : \mathrm{T} \rightarrow \mathrm{C}$, which generates a solution code C for a given visual programming task T. To evaluate $f$ on a task T, we first use $f$ to synthesize a code $\hat{\mathrm{C}}$ and then execute the synthesized code $\hat{\mathrm{C}}$ using an emulator. The emulator outputs *success* if the synthesized code $\hat{\mathrm{C}}$ successfully solves the task T and adheres to code constraints; otherwise, the emulator outputs *fail*. We use *success* as the main evaluation metric. Given a dataset $\mathcal{D}_{\mathrm{eval}} = \{\mathrm{T}_i\}_{i=1}^{N}$, we calculate the success rate of $f$ on this dataset as the overall performance. We curate a dataset REAL of $N = 85$ real-world visual programming tasks from XLogoOnline-Mini and we use this as one of the main datasets for evaluation. In Figures 3a and 3b, we show the overall distribution of this dataset and the number of tasks requiring specific skills, respectively.

# 4 METHODOLOGY FOR SYNTHETIC DATASET GENERATION AND FINE-TUNING

As discussed in Section 1, existing large models such as GPT-4V and Llama3-70B struggle with visual programming tasks in XLogoOnline-Mini. To address this, we develop a two-stage fine-tuning pipeline consisting of synthetic dataset generation and supervised fine-tuning. This section details the dataset generation process and the methodology for fine-tuning large models on the synthetic dataset.

## 4.1 SYNTHETIC DATASET GENERATION

Our goal is to develop a large synthetic dataset for training models (Bunel et al., 2018). To achieve this, we adopt the task synthesis techniques from (Ahmed et al., 2020; Wen et al., 2024), which were developed to automatically generate high-quality tasks in visual programming domains. Instead of random task generation, these techniques allow us to perform more controlled and systematic task

| Task Type | # | Code Constraints | # | Code Concepts | # | Code Length | # | Grid Size | # |
|---|---|---|---|---|---|---|---|---|---|
| Find | 36,055 | None | 34,680 | Basic Actions | 53,779 | Short (1-5) | 20,985 | $Size \leq 3$ | 35,908 |
| Draw | 24,851 | AtMost | 29,354 | Loops | 24,432 | Medium (6-10) | 45,682 | $Size = 4$ | 25,933 |
| Math | 14,994 | Exactly | 16,169 | Variables | 5,931 | Long (11-17) | 22,386 | $Size = 5$ | 14,852 |
| Logic | 13,153 | StartBy | 1,430 | Loops and Variables | 4,911 | | | $Size = 6$ | 8,061 |
| | | Hybrid | 7,420 | | | | | $Size \geq 7$ | 4,299 |
| Total | 89,053 | Total | 89,053 | Total | 89,053 | Total | 89,053 | Total | 89,053 |

(a) Task distribution of SIM dataset.

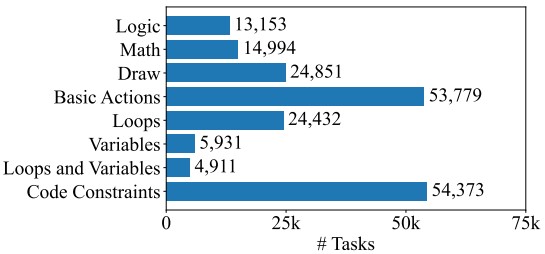

| SIM dataset | # |
|---|---|
| Train Size | 87,053 |
| Validation Size | 1,000 |
| Evaluation Size | 1,000 |
| Total Size | 89,053 |

(b) Skill distribution of SIM dataset.                    (c) Dataset split of SIM dataset.

Figure 4: Statistics of the synthetic SIM dataset. (a) and (b) show the task distribution and the skill distribution, respectively. (c) shows the dataset split.

synthesis, such as specifying task types, code concepts, and code lengths, enabling us to generate tasks with different skills and difficulty levels.

**Adapting task synthesis techniques.** Given a task-code pair as a reference input, the original task synthesis techniques can produce a small, predefined number of tasks and solution codes suited for educational purposes (Ahmed et al., 2020; Wen et al., 2024). Since our goal is to develop a large and diverse dataset for training, we make two key modifications: (i) we remove scoring functions, enabling us to generate a large quantity of tasks instead of a limited selection for educational uses; (ii) we relax task synthesis parameters to enhance techniques' ability to generate more tasks, including allowing larger grid sizes and longer code lengths. While the resulting tasks may not be ideal for educational purposes, they are diverse and challenging for training large models.

**Dataset generation process and statistics.** We use the adapted task synthesis technique to generate a synthetic dataset as follows: (i) we manually craft a solution code for each task in the REAL dataset, resulting in a set $\{(\mathtt{T}_i, \mathtt{C}_i)\}_{i=1}^{85}$; (ii) for each $(\mathtt{T}_i, \mathtt{C}_i)$, we generate up to $1,500$ synthesized tasks and their solution codes. To ensure the quality of the dataset, we take the following processing steps: we remove any duplicate task-code pairs to maintain diversity, conduct a correctness check on the generated solution codes using the emulator, and exclude any task-code pairs present in the real-world REAL dataset from our synthetic dataset. This last processing step guarantees that the model has not seen any tasks from the evaluation dataset during training. We ultimately produce the synthetic dataset SIM with $89,053$ task-code pairs. The statistics of this dataset are detailed in Figure 4. Note that the distribution of this synthetic dataset slightly differs from the real-world dataset REAL (see Figure 3a) due to the aforementioned processing steps and the fact that not all reference tasks can generate the desired number of synthesized tasks. From this synthetic dataset, we randomly select $1,000$ samples for validation, $1,000$ samples for evaluation, and the remaining samples for training. We use this synthetic evaluation dataset ($1,000$ samples), referred to as SIM-EVAL, to complement the real-world dataset REAL in evaluating the model's performance. We provide full details of the dataset generation process and dataset quality assessment in the supplementary material.

### 4.2 METHODOLOGY FOR FINE-TUNING

**Translating tasks and codes.** In our synthetic dataset, tasks and codes are represented in JSON format for ease of parsing and interpretation. However, directly using the JSON format can be challenging for training large models, which are typically pre-trained on natural language texts. Therefore, we translate the JSON representations of each task and code into natural language descriptions and Python-style code, respectively, using a fixed template shown in Figure 5a.

**Supervised fine-tuning using synthetic dataset.** Fine-tuning can involve adjusting all model parameters, modifying only a few layers, or adding new layers (Han et al., 2024). However, fully

You are presented with a visual programming task involving a goal, a grid, a turtle, various items (or lines). You need to write Python code that enables the turtle to accomplish the goal within the grid.

{description_of_grid_properties}
{description_of_python_functions}

Now, write a correct Python code that successfully solves the following task.
### Task:
{description_of_task}
### Goal:
{description_of_goal}
### Correct code:

(a) Prompt template.

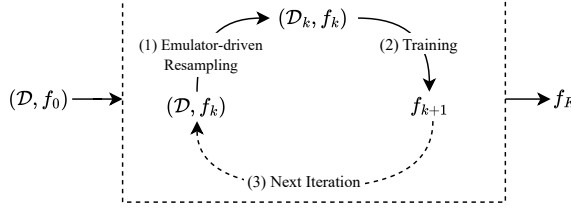

(b) Overview of emulator-driven fine-tuning.

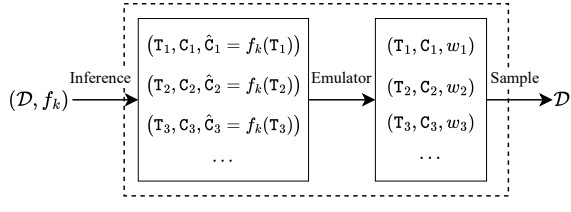

(c) Emulator-driven resampling.

Figure 5: (a) shows the prompt template for fine-tuning. This prompt has several placeholders to include details for the descriptions of different aspects of the task. More details can be found in the supplementary material. (b) provides an overview of emulator-driven fine-tuning, starting with the dataset $\mathcal{D}$ and initial model $f_0$, and iteratively resampling and training to produce the final model $f_K$. (c) illustrates the resampling process in emulator-driven fine-tuning to create the dataset $\mathcal{D}_k$.

fine-tuning all parameters can be computationally expensive and time-consuming. Therefore, we adopt Low-Rank Adaptation (LoRA) (Hu et al., 2022), a parameter efficient fine-tuning technique which introduces trainable rank decomposition matrices into the model's network weights. We train models on the SIM dataset using LoRA in a standard supervised manner. The model receives a natural language task description as input and outputs Python-style code. The model is optimized to minimize the cross-entropy loss between the predicted code and the ground truth solution code.

**Emulator-driven feedback for fine-tuning.** Standard supervised fine-tuning assigns equal weights to all samples in the training dataset. However, our domain presents a unique challenge where tasks vary widely in required skills and difficulty levels (see Figure 4). Additionally, some skills serve as prerequisites for mastering more advanced ones. For instance, a model typically needs to understand basic actions before mastering loops and variables, and it generally solves tasks with shorter code lengths before being able to tackle longer ones. Consequently, treating all tasks with equal importance can be suboptimal in our setting (Bengio et al., 2009). To address this, we introduce emulator-driven fine-tuning, which designs a curriculum over training data distribution by leveraging emulator feedback. The key idea is to dynamically adjust the training data distribution based on the emulator's evaluation of the model, assigning higher weights to tasks where the model struggles, thereby progressively guiding the model from simpler tasks that it can easily solve to more complex tasks.

The overall process is shown in Figure 5b and 5c. More formally, given an initial model $f_0$ and the training dataset $\mathcal{D}$, our goal is to learn a final model $f_K$. To achieve this, at each training epoch $k$, we first perform the *emulator-driven resampling* step (see Figure 5c), where we use the model $f_k$ to infer on the training dataset $\mathcal{D}$ to obtain the predicted code $\hat{C}_i$ for each task $T_i$. We evaluate each predicted code using an emulator and update the weight $w_i$ for $(T_i, C_i)$ as follows:

$$w_i = \frac{1}{|\mathcal{D}|}\big[1 + \beta \cdot \mathbb{I}\big(\text{Emulator}(T_i, \hat{C}_i) = \textit{fail}\big)\big], \tag{1}$$

where $\mathbb{I}(\cdot)$ is an indicator function that returns 1 if the predicted code fails to solve $T_i$, and 0 otherwise. The hyperparameter $\beta$ is adjustable, with a larger $\beta$ encouraging the model to focus more on its mistakes and $\beta = 0$ equivalent to fine-tuning without resampling. Then, we sample the training dataset $\mathcal{D}$ according to the categorical distribution $w_i' = w_i / \sum_{j=1}^{|\mathcal{D}|} w_j$, obtaining a resampled dataset $\mathcal{D}_k$. After resampling, we perform the *training* step, where we train the model $f_k$ using the resampled dataset $\mathcal{D}_k$ to obtain the model $f_{k+1}$. Finally, we repeat the resampling and training steps until the model converges or reaches a predefined number of training epochs, yielding the final model $f_K$. To reduce computational costs, resampling can be performed at fixed intervals (set to 3 epochs in our experiments).

# 5 EXPERIMENTAL EVALUATIONS

In this section, we evaluate the performance of large and fine-tuned models on the XLOGOMINIPROG benchmark. We first outline the experimental setup in Section 5.1, then present the main results and failure analysis in Sections 5.2 and 5.3, followed by additional analysis in Section 5.4.

## 5.1 EXPERIMENTAL SETUP

**Models evaluated.** We compare a range of large models and their fine-tuned versions. All models are queried with temperature 0. We evaluate the following models:

- *Large language models (LLMs).* We evaluate the following LLMs: (i) GPT-3.5 model (version gpt-3.5-turbo-0125) (OpenAI, 2023a); (ii) GPT-4 model (version gpt-4-turbo-2024-04-09) (OpenAI, 2023b); (iii) Llama2 and Llama3 models with 7B, 13B, and 70B parameters, respectively (version instruction-tuned) (Touvron et al., 2023; Meta, 2024).

- *Vision language models (VLMs).* We evaluate the following VLMs: (i) GPT-4V model (version gpt-4-turbo-2024-04-09); (ii) Llava1.5 models (version llava-v1.5-7B and llava-v1.5-13B) (Liu et al., 2023a); (iii) InternVL2 models (versions InternVL2-8B and InternVL2-Llama3-76B) (Chen et al., 2023); (iv) Qwen2VL models (versions Qwen2VL-7B-Instruct and Qwen2VL-72B-Instruct) (Wang et al., 2024); (v) NVLM-D model (version NVLM-D-72B) (Dai et al., 2024); and (vi) Molmo models (versions Molmo-7B-D and Molmo-72B) (Deitke et al., 2024). VLMs are queried in the same way as LLMs, but with a task image provided as additional input to leverage their vision capabilities.

- *Fine-tuned models.* We fine-tune the Llama2-7B, Llama3-8B, and Llava1.5-13B models using our synthetic dataset. Llama3-8B-Uni is fine-tuned on our synthetic training dataset with uniform data distribution (i.e., standard fine-tuning). Llama3-8B-Emu is fine-tuned on the same dataset with emulator-driven resampling in Section 4.2. We apply the same fine-tuning procedures to Llama2-7B base models, yielding Llama2-7B-Uni and Llama2-7B-Emu. For Llava1.5-13B, we apply standard supervised fine-tuning with task images, resulting in Llava1.5-13B-Uni. Additional fine-tuning details are in the supplementary material.

**Evaluation procedure and metrics.** We evaluate models using two datasets: REAL and the synthetic dataset SIM-EVAL (see Figure 4c). For each task in our evaluation datasets, we first convert the task from JSON format into natural language description using a fixed prompt template (see Figure 5a).[1] For multimodal models (e.g., GPT-4V, Llava1.5), we also provide an image of the task as additional input to the model. Then, we use the model to generate the code in Python programming language. However, the model might produce the natural language explanation alongside code. We extract only the Python code from the models' outputs. Finally, we run the extracted code using an emulator and evaluate the model. We use *success* as the main metric (see Section 3.2), and also consider two additional metrics: (i) *Format*, which evaluates if the model's output adheres to the desired code format, and (ii) *No-Crash*, which evaluates if the code runs without crashing, such as hitting walls, entering forbidden areas, or exceeding grid boundaries.

## 5.2 MAIN RESULTS

**Base models' performance.** The results are shown in Figure 6. Among the base models evaluated, GPT-4V performs the best with a success rate of $20.00\%$ on the REAL dataset. Notably, GPT-4V outperforms GPT-4, which has a success rate of $12.94\%$. This suggests that incorporating visual information can enhance the performance of large models on visual programming tasks. However, all other base models, including GPT-3.5, Llama, and Llava models, perform poorly on REAL. Regarding the synthetic evaluation dataset SIM-EVAL, we find that the performance of most base models declines. This is because the tasks in SIM-EVAL are more challenging than those in REAL in terms of code length and grid size (see Figure 3 and 4). For example, the calculated percentage of code with length "Long (11-17)" in SIM-EVAL is $25.14\%$, compared to $17.65\%$ in REAL.

---

[1]The prompt template does not include few-shot examples or advanced prompting strategies. The evaluation of different prompting strategies is provided in the supplementary material.

| | REAL (85 samples) | | | SIM-EVAL (1,000 samples) | | |
|---|---|---|---|---|---|---|
| | Format (%) | No-Crash (%) | Success (%) | Format (%) | No-Crash (%) | Success (%) |
| **Base LLMs** | | | | | | |
| GPT-3.5 | 92.94 | 11.76 | 1.18 | 87.60 | 9.50 | 1.60 |
| GPT-4 | **95.29** | **38.83** | **12.94** | **97.40** | **16.80** | **5.30** |
| Llama3-8B | 48.24 | 5.88 | 0.00 | 40.90 | 2.80 | 0.60 |
| Llama3-70B | 67.06 | 8.24 | 2.35 | 15.50 | 1.20 | 0.30 |
| Llama2-7B | 27.06 | 5.88 | 0.00 | 21.90 | 2.90 | 0.40 |
| Llama2-13B | 60.00 | 7.06 | 0.00 | 54.40 | 3.50 | 0.40 |
| Llama2-70B | 28.24 | 7.06 | 0.00 | 38.30 | 1.10 | 0.10 |
| **Base VLMs** | | | | | | |
| GPT-4V (Vision) | **96.47** | **47.06** | **20.00** | **95.50** | **18.10** | **5.50** |
| Llava1.5-7B | 10.59 | 1.18 | 0.00 | 3.20 | 0.00 | 0.00 |
| Llava1.5-13B | 10.59 | 2.35 | 0.00 | 9.00 | 2.10 | 0.00 |
| InternVL2-8B | 0.00 | 0.00 | 0.00 | 56.90 | 3.80 | 0.00 |
| InternVL2-Llama3-76B | 77.65 | 31.76 | 9.41 | 40.50 | 6.10 | 1.50 |
| Qwen2VL-7B-Instruct | 43.53 | 9.41 | 0.00 | 14.30 | 2.10 | 0.20 |
| Qwen2VL-72B-Instruct | 28.24 | 11.76 | 0.00 | 36.50 | 4.40 | 0.40 |
| NVLM-D-72B | 61.18 | 8.24 | 1.18 | 67.40 | 8.30 | 2.00 |
| Molmo-7B-D | 75.29 | 8.24 | 0.00 | 66.00 | 7.70 | 0.60 |
| Molmo-72B | 4.71 | 1.18 | 1.18 | 6.40 | 0.70 | 0.40 |
| **Fine-tuned models** | | | | | | |
| Llava1.5-13B-Uni | 68.24 ± 18.48 | 19.53 ± 14.98 | 11.99 ± 10.55 | 56.18 ± 15.68 | 13.64 ± 11.36 | 10.68 ± 10.23 |
| Llama2-7B-Uni | 99.76 ± 0.24 | 65.88 ± 1.05 | 45.65 ± 0.86 | 99.98 ± 0.02 | 62.64 ± 0.33 | 53.04 ± 0.20 |
| Llama2-7B-Emu | **100** ± 0.00 | 69.41 ± 1.97 | 51.53 ± 0.44 | 99.96 ± 0.02 | 68.70 ± 0.49 | 60.10 ± 0.69 |
| Llama3-8B-Uni | 99.53 ± 0.29 | **73.65** ± 0.80 | 54.12 ± 1.78 | 99.96 ± 0.04 | 71.26 ± 1.01 | 62.72 ± 1.17 |
| Llama3-8B-Emu | 99.76 ± 0.24 | 71.53 ± 0.78 | **60.23** ± 1.01 | **100** ± 0.00 | **74.92** ± 0.60 | **66.92** ± 0.65 |

Figure 6: Performance comparison of models on two evaluation datasets. Bold values indicate the highest performance in each column across base or fine-tuned models. Fine-tuned models are trained using 5 different random seeds and we report the mean and standard error of the performance.

**Effectiveness of fine-tuning.** Standard fine-tuning on a domain-specific dataset enhances the performance of base models, especially Llama models. As shown in Figure 6, after standard fine-tuning, the success rate for Llama3-8B-Uni is 54.12% on REAL and 62.72% on SIM-EVAL. Similar improvements are observed for Llama2-7B-Uni. However, the improvement for Llava1.5-13B-Uni does not match the gains from fine-tuning the Llama models, and exhibits inconsistent performance across different seeds, as shown by the large standard errors. We also note that the performance of fine-tuned models on REAL generally lags behind their performance on SIM-EVAL. This is because the task distribution of SIM-EVAL more closely resembles the training dataset due to the dataset split. Our results also show that emulator-driven resampling effectively enhances fine-tuning performance. Llama3-8B-Emu achieves a success rate of 60.23% and 66.92% on REAL and SIM-EVAL, respectively, outperforming Llama3-8B-Uni by 6.11% and 4.20%.[2]

## 5.3 FAILURE ANALYSIS

In this section, we perform failure analysis to better understand the limitations of different models. We conduct two types of failure analysis: (i) *explanation-based failure analysis*, where we examine the explanations generated by the models to identify the reasons for failures, and (ii) *perturbation-based failure analysis*, where we evaluate the models' performance on simplified, perturbed tasks.

**Explanation-based failure analysis.** We first present a failure analysis by analyzing the output codes and explanations of different models. We consider base models, specifically GPT-4V and Llama3-70B, as fine-tuned models are trained to generate code without explanations. To conduct the failure analysis, we first identify common failure types, which are categorized as follows: (i) *Repetition*: generating the same code sequences repeatedly; (ii) *Format*: producing code with incorrect formatting, including the use of disallowed commands; (iii) *Goal*: misinterpreting the goal or attempting to devise a tricky approach to achieve the goal; (iv) *Code constraints*: failing to adhere to specified code constraints while solving the task; (v) *Grid constraints*: attempting to solve the task while ignoring walls, forbidden cells, or grid boundaries; (vi) *Spatial reasoning*: misunderstanding coordinates or directions following movements or turns; (vii) *Hallucination*: generating non-existent items or code commands. Then we systematically analyze the explanations generated by the models alongside the output code and manually annotate the underlying reasons for each failure. In cases where multiple failure reasons are identified, we attribute the failure to the most significant cause. The results of this analysis are shown in Figure 7a. Our findings indicate that both GPT-4V and Llama3-70B exhibit the most difficulty with spatial reasoning. For Llama3-70B, another primary failure type is the repetition, where the model generates the same code sequences repeatedly.

---

[2]However, fine-tuning on domain-specific datasets can also lead to a performance drop in other domains. Additional results are provided in the supplementary material.

| | Repetition | Format | Goal | Code Constraints | Grid Constraints | Spatial Reasoning | Hallucination | Success |
|---|---|---|---|---|---|---|---|---|
| GPT-4V | 0.00 | 3.53 | 11.76 | 7.06 | 11.76 | **42.35** | 3.53 | 20.00 |
| Llama3-70B | 34.12 | 1.18 | 5.88 | 3.53 | 8.24 | **44.71** | 0.00 | 2.35 |

(a) Failure rates (%) of different failure types by analyzing model outputs on the REAL dataset. Bold values highlight the most common failure type for each model.

| | $T$ | $T_A$ | $T_B$ | $T_C$ | $T_{A,B}$ | $T_{B,C}$ | $T_{A,C}$ | $T_{A,B,C}$ |
|---|---|---|---|---|---|---|---|---|
| GPT-4V | 0.00 | 30.00 | 30.00 | **50.00** | 50.00 | 50.00 | **60.00** | **60.00** |
| Llama3-70B | 0.00 | 0.00 | 0.00 | 0.00 | 0.00 | 0.00 | 0.00 | 0.00 |
| Llama3-8B-Uni | 0.00 | 0.00 | **10.00** | 0.00 | **20.00** | **20.00** | 0.00 | **30.00** |

(b) Success rates (%) of models across 80 perturbed tasks. Each type of perturbation includes 10 tasks. Perturbations are grouped by the number of components removed. Bold values indicate the highest success rate for each model within each perturbation group.

Figure 7: Explanation-based and perturbation-based failure analysis on the REAL dataset. (a) highlights the main reasons for model failures by analyzing model explanation, with spatial reasoning being the primary reason for both GPT-4V and Llama3-70B. (b) presents the success rates of models on perturbed tasks, showing that GPT-4V faces difficulties with spatial reasoning, while the fine-tuned Llama3-8B-Uni struggles most with grid constraints.

**Perturbation-based failure analysis.** We provide another type of failure analysis by perturbing tasks to understand the limitations of different models. In this analysis, we consider both base and fine-tuned models, including GPT-4V, Llama3-70B, and Llama3-8B-Uni. We first select 10 tasks from the REAL dataset that the three models consistently fail to solve. For each task, we consider three types of perturbations: (A) removing code constraints, (B) removing grid constraints (i.e., walls and forbidden cells), and (C) simplifying spatial relationships (i.e., moving the turtle closer to the target). If a task lacks certain components (e.g., no code or grid constraints), we leave the task unchanged. In total, we analyze 80 tasks (10 selected tasks × 8 perturbed versions per task). These 8 perturbed versions include: the original tasks ($T$), 3 tasks with one component removed ($T_A$, $T_B$, $T_C$), 3 tasks with two components removed ($T_{A,B}$, $T_{A,C}$, $T_{B,C}$), and 1 task with all components removed ($T_{A,B,C}$). Finally, we evaluate the performance of different models on these perturbed tasks. As shown in Figure 7b, GPT-4V struggles most with handling spatial relationships. When simplifying spatial relationships, GPT-4V's success rate increases significantly, from $0\%$ to $50.0\%$ (see columns $T$ and $T_C$). On the other hand, Llama3-8B-Uni struggles most with grid constraints. Removing grid constraints improves its success rate to $10.0\%$ (column $T_B$), while removing the other two components (code constraints and spatial relationships) has no noticeable effect on its performance. [3]

## 5.4 ADDITIONAL RESULTS AND ANALYSIS

In this section, we provide additional experiments and results to further analyze the performance of models on our visual programming tasks.

**Comparative analysis of models' capabilities.** We evaluate model performance across various dimensions to identify strengths and weaknesses. To this end, we automatically categorize each task-code pair according to different dimensions (e.g., task type). We determine the model's capability in a specific aspect within a dimension (e.g., Math in the task type dimension) by calculating the success rates for all tasks involving that aspect. In Figure 8, we present a comparative analysis of three representative models—GPT-4V, Llama3-70B, and Llama3-8B-Emu—across distinct dimensions using the REAL dataset. Overall, Llama3-8B-Emu consistently outperforms other models across all dimensions and GPT-4V shows superior performance compared to Llama3-70B in most aspects. In Figure 8a, 8b, and 8c, we find that the GPT-4V and Llama3-70B struggle most with tasks of type "logic" and with scenarios that impose code constraints such as AtMost and Exactly. In terms of code concepts, Llama3-70B fails to solve any tasks that require variables, showing its limitations in handling complex programming concepts in visual programming. Notably, as shown in Figure 8d,

---

[3]Interestingly, our failure analysis shows that Llama3-8B-Uni performs worse than GPT-4V. This may be due to the failure analysis tasks having a different distribution than those in Figure 6. Specifically, we select tasks that all models initially failed on, indicating that Llama3-8B-Uni already struggles with them. After perturbation, these tasks diverge further from the training data distribution. In contrast, GPT-4V is not affected by these distribution shifts and its stronger generalization abilities make it perform better on these tasks.

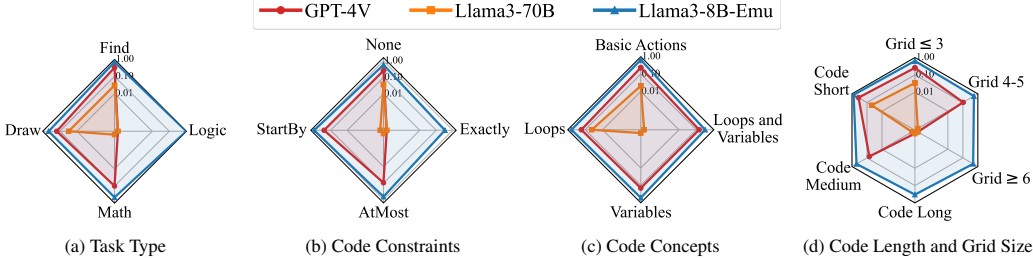

(a) Task Type      (b) Code Constraints      (c) Code Concepts      (d) Code Length and Grid Size

Figure 8: Comparative analysis of models' capabilities across different dimensions on REAL. Each chart highlights the models' capabilities in different aspects within a dimension. Note that code length and grid size are combined in the same chart, as both indicate the difficulty levels of the tasks. The performance metrics are logarithmically scaled to enhance clarity.

the performance of all models declines with increasing difficulty of tasks, as indicated by longer code lengths and larger grid sizes. GPT-4V fails to solve tasks requiring long code sequences or grid sizes larger than 6. Llama3-70B performs even more poorly, starting to fail on tasks requiring medium-length codes and grid sizes larger than 3.

**Can fine-tuned models learn transferable skills?** We explore whether fine-tuned models can develop transferable skills to solve tasks that are not seen during training. To investigate this, we first exclude all tasks involving math skills (e.g., Task 38 in Figure 1) from the training dataset, resulting in a reduced training dataset with 72k samples. Then we fine-tune Llama3-8B on this reduced dataset using standard supervised learning, referring to the resulting model as *Llama3-8B-Uni (no-math)*.

Next, we evaluate this model exclusively on math tasks from the evaluation datasets. The results are shown in Figure 9. Our results reveal that Llama3-8B-Uni (no-math) outperforms Llama3-70B, despite neither model being trained on math tasks. This suggests that the fine-tuned Llama3-8B-Uni (no-math) acquires certain transferable skills. However, compared to Llama3-8B-Uni, which was trained on the

|  | REAL (10 tasks) | SIM-EVAL (176 tasks) |
|---|---|---|
| Llama3-70B | 0.00 | 0.00 |
| Llama3-8B-Uni (no-math) | $10.00 \pm 10.00$ | $6.25 \pm 1.18$ |
| Llama3-8B-Uni | $\mathbf{40.00} \pm 5.48$ | $\mathbf{38.98} \pm 1.82$ |

Figure 9: Success rates (%) of models on math tasks. Success rates of fine-tuned models are reported as mean and standard error across five seeds.

full dataset including math tasks, the no-math version performs much worse. This indicates that while Llama3-8B-Uni (no-math) learns some generalizable skills, it is less effective than the model trained directly on data that includes those skills.

## 6   CONCLUDING DISCUSSIONS

**Summary.** In this paper, we introduced the XLOGOMINIPROG benchmark to evaluate the program synthesis capabilities of large models within the XLogoOnline visual programming environment. We found that large models struggle with visual programming tasks that require a combination of skills, despite our benchmark tasks only requiring basic programming skills. Our best evaluated base model, GPT-4V, only achieved a $20\%$ success rate. To improve performance, we developed a fine-tuning pipeline that involves synthetic dataset generation followed by supervised fine-tuning. This pipeline enabled the Llama3-8B model to achieve a success rate of $54.12\%$ on the benchmark tasks. Additionally, we demonstrated that leveraging emulator-driven feedback can further enhance standard fine-tuning performance by approximately $6\%$ in both Llama3-8B and Llama2-7B models. Through failure analysis, we found that GPT-4V and Llama3-70B struggle most with spatial reasoning, while the fine-tuned Llama3-8B-Uni faces the most difficulty with grid constraints.

**Limitations and future work.** We discuss some limitations of our work and propose ideas for addressing them in the future. First, our benchmark focuses on basic programming skills, and future work could extend it to include more complex programming tasks. This could involve tasks that require more advanced programming concepts, such as conditionals and functions. Second, our emulator-driven fine-tuning provides the model with only binary feedback on the correctness of the predicted code. In the future, it would be interesting to provide more detailed feedback, such as identifying specific errors in the generated code and then using this more informative feedback to guide the fine-tuning process.

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

## A    TABLE OF CONTENTS

In this section, we provide a brief description of the content provided in the appendices of the paper.

- Appendix B provides more details about the datasets.
- Appendix C provides more details about the fine-tuning process and evaluation.
- Appendix D provides additional experiments and results.
- Appendix E provides more details about the prompts used for fine-tuning and evaluation.

## B    MORE DETAILS ABOUT THE DATASETS

We provide the following details about the datasets.

1. *Real-world tasks in the XLogoOnline platform:* The real-world visual programming tasks in the REAL dataset are curated from the Mini level of the XLogoOnline platform. These real-world programming tasks can be accessed and viewed at `https://xlogo.inf.ethz.ch/`. Figure 10 shows the screenshots of the platform.

2. *The benchmark source code and datasets:* See the provided file **iclr2025-xlogo-benchmark_src.zip**. After unzipping this file, the dataset is available in the *data.zip* file.

3. *Data license confirmation:* We confirm that all data used in this paper is either publicly available or has been obtained and used in accordance with the relevant data licenses.

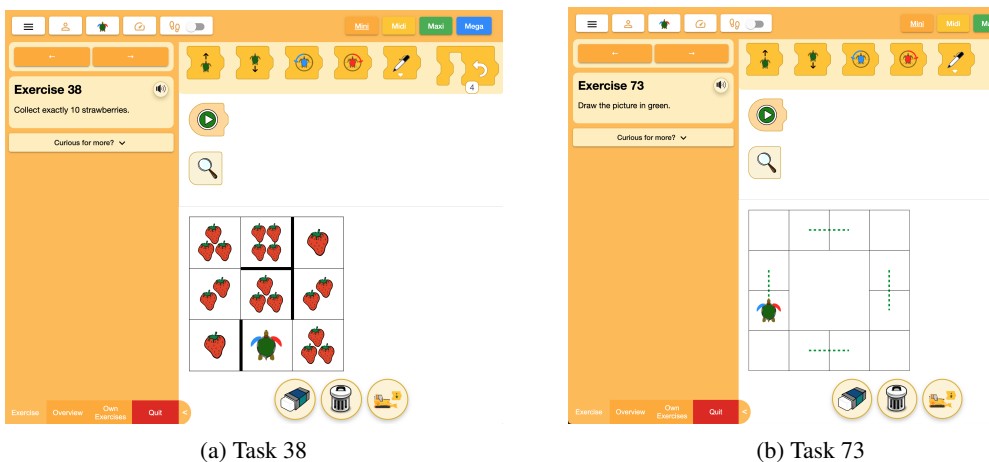

(a) Task 38                                           (b) Task 73

Figure 10: Example tasks from the XLogoOnline platform. Students need to drag and drop different blocks to solve the tasks.

### B.1    DETAILS OF THE SYNTHETIC DATASET GENERATION

In this section, we provide more details about the generation process of the synthetic dataset SIM.

We use the adapted task synthesis technique (Ahmed et al., 2020; Wen et al., 2024) to generate a synthetic dataset. The key idea is to take a reference task and its solution code as input, and then apply symbolic execution and constraint satisfaction techniques to systematically enumerate all possible task-code outputs. The details are described as follows.

First, we manually craft a solution code for each of the $N = 85$ tasks in the REAL dataset, resulting in a set $\{(\mathtt{T}_i, \mathtt{C}_i)\}_{i=1}^N$. However, our objective is to generate a large and diverse set of tasks to train large models. To achieve this, we specify an additional parameter difficulty level $\mathtt{D}$. This parameter enables us to generate tasks with varying levels of difficulty by specifying the desired code length, number of code constraints, and goals relative to the reference input, thereby enhancing the diversity of the dataset. The parameters are detailed as follows:

- `Easy`: The code length and number of code constraints remain the same as in the reference code and code constraints, and the goal remains unchanged.
- `Medium`: The code length is increased by 1 or 2 additional commands compared to the reference code, while the number of code constraints and the goal remain the same as in the reference task `T`.
- `Hard`: The code length is increased by up to 2 additional commands, one more code constraint is added compared to the reference code constraints, and the goal may be modified.

Note that the difficulty levels mentioned above indicate the relative difficulty of the generated tasks compared to the reference task, not the absolute difficulty of the tasks.

Given the reference input $(T, C, D)$, we begin by enumerating all possible codes, code constraints, and goals that meet the specified difficulty levels. To achieve this, we first create templates for the code, constraints, and goals, respectively, each containing placeholders. These placeholders are then populated with specific values using an SMT-based constraint solver (de Moura & Bjørner, 2008). This process allows us to generate all possible combinations of code, constraints, and goals that align with the desired difficulty levels.

Next, we generate task-code pairs by combining the previously generated code, code constraints, and goals with corresponding grid worlds. To generate these grid worlds, we symbolically execute the previously generated code within an empty grid, constructing elements like walls and target items to ensure the grid can be successfully solved by the code. After the grid world is constructed, it is combined with the corresponding code, code constraints, and goal to form a task-code pair.

In implementation, we generate up to 3,000 tasks for each combination of code, code constraints, and goals. Subsequently, we sample 500 tasks from the pool of all generated tasks for each $(T, C, D)$, resulting in up to 500 tasks $\times$ 3 difficulty levels $= 1,500$ tasks for each reference input $(T, C)$. This process is repeated for all reference inputs in the dataset, resulting in a total of up to $85 \times 1,500 = 127,500$ tasks. Finally, we apply the processing steps described in the main paper to generate the synthetic dataset, resulting in the final dataset, SIM, containing $89,053$ tasks and solution codes.

To run the adapted task synthesis technique, we use a 12-core, 3 GHz Intel Xeon E7-8857 CPU, with parallelization across 8 cores under a 64-bit Debian operating system.

## B.2 Quality of the Datasets

The quality of the datasets is crucial for the success of the models trained on them. Therefore, we provide the more details about the quality of the datasets. We mainly use the following two datasets for evaluation:

1. REAL dataset (85 samples): This dataset was derived from the visual programming platform XLogoOnline. The tasks included in this platform were meticulously crafted by experts and have been used by tens of thousands of students every year (Hromkovic et al., 2017; Staub, 2021). Given this extensive use and expert involvement, the quality of the tasks in this dataset is guaranteed.

2. SIM-EVAL dataset (1000 samples): This dataset was synthetically generated. However, we ensure data quality by implementing the following checks: (i) we have removed any duplicate task-code pairs; (ii) we have conducted a correctness check on the generated solution codes using the emulator, and (iii) we have excluded any task-code pairs present in the REAL dataset from this synthetic dataset. In Figure 12, we show examples of the tasks in this dataset.

To further demonstrate the quality of our datasets, we conduct a quality annotation for both datasets. Specifically, we annotate the quality of all 85 samples in the REAL dataset and randomly sample 5% of tasks from the SIM-EVAL dataset for annotation. The following rubrics are used to evaluate each (task, code) pair:

1. *Visual appeal*
   - 0: Poor - The visual grid is highly unappealing.
   - 0.5: Acceptable - The visual grid is moderately appealing.

- 1: Excellent - The visual grid is highly appealing.

2. *Grid elements utility*
    - 0: Poor - The distractors are neither useful nor reasonably positioned.
    - 0.5: Acceptable - Some distractors are useful, while others lack utility.
    - 1: Excellent - Most, if not all, distractors are strategically useful and sensibly placed.

3. *Code quality*
    - 0: Poor - The code is of poor quality, unable to solve the task, or violates code constraints.
    - 0.5: Acceptable - The code can solve the task but contains some unnecessary commands.
    - 1: Excellent - The code solves the task, meets code constraints, and has no redundant commands.

4. *Overall quality*: Calculated as the minimum score across visual appeal, grid elements utility, and code quality.

|           | Visual Appeal | Grid Elements Utility | Code Quality | Overall Quality |
|-----------|---------------|-----------------------|--------------|-----------------|
| REAL      | 1.00          | 1.00                  | 1.00         | 1.00            |
| SIM-EVAL  | 0.97          | 0.94                  | 0.89         | 0.84            |

Figure 11: Quality annotation results for REAL and SIM-EVAL datasets. For REAL, we annotate all 85 samples, while for SIM-EVAL, we randomly sample 5% of the dataset for annotation.

The results in Figure 11 demonstrate that the overall quality of the REAL dataset is excellent. The SIM-EVAL dataset, with an overall quality score of $0.84$, exceeds the acceptable threshold (score = $0.5$) and approaches the level of excellence (score = $1.0$). Additionally, during the quality annotation, we do not find any (task, code) pair where the task is unsolvable or the code fails to successfully solve the task.

## C  MORE DETAILS OF THE FINE-TUNING AND EVALUATION

**Details of fine-tuning Llama family models.** For Llama family models, we choose non-instruction-tuned versions for fine-tuning because the base models will be fine-tuned to generate code, without requiring instruction-following capabilities. We use LoRA for parameter-efficient fine-tuning (Hu et al., 2022). To find the best LoRA rank and scaling factor, we experimented with ranks of 8, 16, 32, and 64, using a scaling factor $\alpha$ four times the rank in each case. We found that a rank of 32 and 64 provide the best performance. Consequently, we use a rank of 32 and a scaling factor of 128 for all fine-tuning experiments. Fine-tuning is performed with a batch size of 4 and a learning rate of $1 \times 10^{-4}$. All fine-tuning experiments are conducted on an internal cluster using 4 A100 GPUs. Each epoch of fine-tuning for the Llama3-8B and Llama2-7B models takes approximately 3.75 hours. In our experiments, all fine-tuned Llama models are trained for 8 epochs, as we observed that the validation dataset loss stabilizes around epoch 8. We train all fine-tuned Llama models using 5 different random seeds.

**Details of fine-tuning Llava family model.** We perform standard supervised fine-tuning to Llava1.5-13B (Liu et al., 2023a). To do this, we follow the default fine-tuning setup and code provided by the authors.[4] Specifically, we use LoRA with a rank of 128 and a scaling factor of 256 for fine-tuning Llava1.5-13B. During fine-tuning, we use a batch size of 16, a learning rate of $2 \times 10^{-4}$, and a maximum sequence length of 2048. We fine-tune the Llava model for 3 epochs on the 87k training dataset using 5 different random seeds, utilizing 4 A100 GPUs.

**Details of emulator-driven fine-tuning.** For emulator-driven fine-tuning, we use the same hyperparameters and setup as the standard fine-tuning, with the exception of resampling every 3 epochs. Specifically, we resample the training dataset based on the emulator's evaluation results every 3

---

[4]https://github.com/haotian-liu/LLaVA

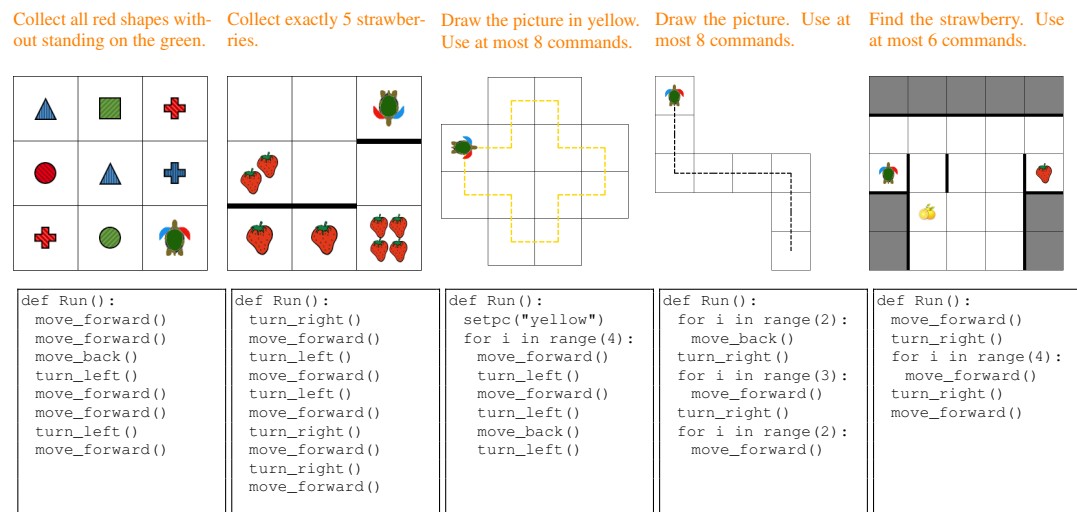

Figure 12: Examples of synthetic tasks and their corresponding solution codes in SIM-EVAL. Note that while the synthesized solution codes are correct, they may not use the minimum number of commands.

epochs. To save time and resources, we start from the checkpoint of the fine-tuned models without resampling at epoch 3. We then reuse this checkpoint to continue fine-tuning for 5 additional epochs using emulator-driven resampling, resulting in a total of 8 epochs. Emulator-driven resampling requires calculating a weight for each training sample, which involves inference over the entire training dataset. For inference, we use the vLLM inference engine (Kwon et al., 2023) with `max_num_seqs` of 8, batch size of 2, and temperature of 0. In this setting, a single iteration of inference and resampling on the 87k training dataset takes approximately 8 hours. After inference, we use the emulator to evaluate the correctness of the model's predicted code. Based on this evaluation, we calculate the weight for each training sample using a value of $\beta = 1$.

**Details of evaluation.** To evaluate GPT family models, we use the OpenAI API with a temperature of 0. For Llama3-8B, Llama2-7B, and fine-tuned models, we use the vLLM (Kwon et al., 2023) inference engine with 2 A100 GPUs, using a temperature of 0 and `max_num_seqs` of 2. We find that a smaller `max_num_seqs` value slows down inference speed but improves performance. Therefore, we choose a `max_num_seqs` value of 2 to balance performance and speed for inference. After inference, we use the emulator to evaluate the models' success rates over the evaluation datasets.

**Details of the emulator.** We have implemented an emulator that can be used to run the code for a given task and provide detailed execution results. The emulator operates in the following way: given a (task, code) pair in our domain, the emulator runs the code for the task and then returns the execution result. During execution, the emulator checks the code format, whether the code execution crashes, whether the code constraints are satisfied, and whether the code can achieve the task's goal. Note that the code constraints and the task's goal are specified in JSON format for precise and unambiguous checking. When creating prompts for models to generate the code, these code constraints and goals are translated into natural language using a fixed translation template. For example, a task's code constraints and goal might be translated as, "Find the strawberry using at most 8 commands." After all above checks are performed, the emulator provides the execution result, which is either "success" or an error message indicating the specific reason for the failure. For example, when code execution is successful for a task, the execution result is "success." If there is an error, such as "hitting the wall," the emulator generates the appropriate error type and message. We use the emulator to evaluate the success rates of the models over the evaluation datasets and also use it to implement our emulator-driven fine-tuning.

|        | Vanilla | 3-shot | 3-shot + CoT |
|--------|---------|--------|--------------|
| GPT-4  | 12.94   | 10.59  | 18.82        |
| GPT-4V | 20      | 14.12  | 15.29        |

Figure 13: Success rates (%) of GPT-4 and GPT-4V with different prompting strategies on the REAL dataset. 3-shot prompting is not notably effective, but when combined with CoT, it leads to performance improvements. However, for GPT-4V, the vanilla prompt is the most effective.

|                   | Success Rates (%) | |
|-------------------|-------------------|--------------------|
|                   | NL                | ASCII              |
| **Base models**   |                   |                    |
| GPT-4             | **12.94**         | 5.88               |
| Llama3-70B        | **2.35**          | 1.18               |
| **Fine-tuned models** |               |                    |
| Llama3-8B-Uni     | **54.12** ± 1.78  | 53.18 ± 1.01       |

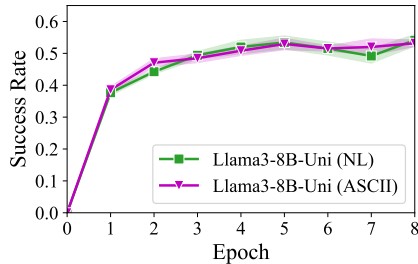

(a) Performance of base and fine-tuned models with NL and ASCII prompts.

(b) Performance of Llama3-8B-Uni across epochs with NL and ASCII prompts.

Figure 14: Influence of task representations on model performance. We compare the performance of base models and fine-tuned models using natural language (NL) and ASCII prompts, respectively. (a) shows the success rates of base and fine-tuned models. (b) shows the performance of fine-tuned models across different epochs. Natural language prompts lead to better performance in base models. However, the fine-tuned Llama3-8B-Uni performs similarly with both NL and ASCII prompts.

# D  ADDITIONAL EXPERIMENTS AND RESULTS

In this section, we present additional experiments and results. First, we investigate the influence of different prompting strategies on model performance. Next, we investigate task representations, comparing natural language and ASCII-based prompts. Then, we analyze the performance of fine-tuned Llama models across different epochs. Finally, we present a case study on output code analysis for perturbed tasks, providing further insights into failure analysis.

## D.1  INFLUENCE OF THE PROMPTING STRATEGIES

Carefully designed prompts have been shown to improve the performance of large models (Wei et al., 2022; Brown et al., 2020). We conduct experiments on different prompting strategies to investigate their effectiveness in our benchmark. We consider the following prompting strategies: (i) *Vanilla* is the prompt without any additional examples or chain-of-thoughts; (ii) *3-shot* is the prompt with 3-shot examples (Brown et al., 2020). (iii) *3-shot + CoT* is the prompt with the 3-shot examples and a step-by-step chain-of-thought (CoT) for each example (Wei et al., 2022). Note that the 3-shot examples are manually designed to ensure they cover most skills, including math, logic, draw, basic actions, variables, loops, and code constraints. These same 3-shot examples are used to prompt all tasks for *3-shot* and *3-shot + CoT* prompting.

The results are shown in Figure 13. We observe that *3-shot* prompting by itself is not notably effective. However, when combined with CoT, it leads to performance improvements, though these gains are limited. We speculate that this is due to the nature of our visual programming tasks, which require long-term path planning, an understanding of spatial relationships, and accurate prediction of the consequences of actions. These elements are typically absent from the training data, making it difficult for the model to leverage in-context learning to solve unfamiliar visual programming tasks.

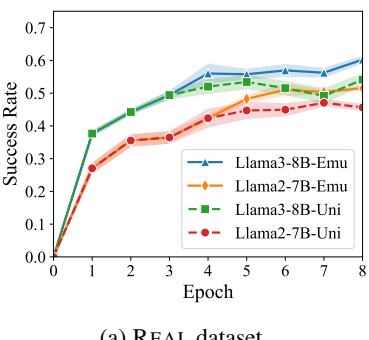 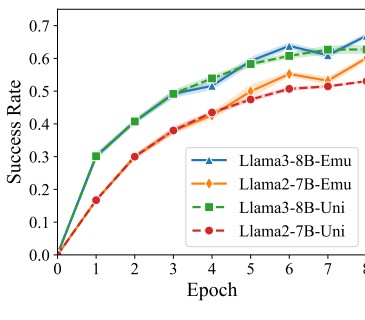

(a) REAL dataset.    (b) SIM-EVAL dataset.

Figure 15: Fine-tuning performance across different epochs on two evaluation datasets. (a) shows the performance of fine-tuned models across different epochs on the evaluation dataset REAL. (b) shows the fine-tuning performance across different epochs on the synthetic evaluation dataset SIM-EVAL.

### D.2 INFLUENCE OF TASK REPRESENTATIONS

In this section, we investigate the influence of natural language and ASCII representations on model performance.

For visual programming tasks, the 2-dimensional grid can be represented in various ways, including natural language descriptions, ASCII-based representations, and images. For the ASCII representation, we developed a template to represent the task's visual grid using ASCII characters. These ASCII characters are then provided to the model as a replacement for the natural language descriptions of the visual grid, both for fine-tuning and evaluation. An example of an ASCII-based prompt is shown in Figure 19.

The evaluation results are shown in Figure 14. Our results indicate that GPT-4 and Llama3-70B perform better with natural language (NL) representations. This might be due to their predominant training on natural language data. However, the fine-tuned Llama3-8B-Uni model performs similarly with both NL and ASCII prompts, with final success rates of $54.12\%$ and $53.18\%$, respectively.

In Figure 14b, we show Llama3-8B-Uni's performance across different epochs with NL and ASCII prompts. We find that the performance of Llama3-8B-Uni with NL and ASCII prompts converges at a similar rate, suggesting that fine-tuning helps the model adapt to ASCII-based task representations, making task representations less critical for fine-tuning models in our visual programming domain.

### D.3 FINE-TUNING PERFORMANCE ACROSS DIFFERENT EPOCHS.

Figure 15a illustrates the performance of fine-tuned models across different epochs. For the emulator-driven fine-tuning (Emu), we adjust the resampling interval to every three epochs, specifically at epochs 3 and 6. At epoch 3, we reuse the checkpoint from the standard fine-tuning (Uni) to save time and resources. As a result, the performance of the emulator-driven fine-tuning (Emu) matches that of the corresponding standard fine-tuning (Uni) up until epoch 3. Then, an emulator-driven resampling is performed at epoch 3, leading to further performance improvements compared to models without resampling. Notably, at the end of training, Llama2-7B-Emu achieves performance close to that of Llama3-8B-Uni, despite the latter being fine-tuned on a more advanced base model. This demonstrates the effectiveness of the curriculum designed by emulator-driven resampling in enhancing the performance of standard fine-tuning.

In Figure 15b, we show the fine-tuning performance across different epochs on the synthetic evaluation dataset SIM-EVAL. This synthetic evaluation dataset exhibits the same distribution as the training dataset due to our splitting method. Emulator-driven resampling is performed at epochs 3 and 6 for both Llama3-8B-Emu and Llama2-7B-Emu. We find that standard fine-tuning without resampling leads to a smooth increase in performance across epochs, as seen in the Llama3-8B-Uni and Llama2-7B-Uni curves. In contrast, emulator-driven fine-tuning with resampling shows slight performance fluctuations, particularly in the epochs immediately following resampling (i.e., epochs 4 and 7). The fluctuations in emulator-driven fine-tuning might be due to the resampling process altering the

|  | HumanEval | HumanEval+ | MBPP | MBPP+ |
|---|---|---|---|---|
| Llama3-8B (Base) | 36.6% | 31.1% | 62.4% | 52.6% |
| Llama3-8B-Uni (Fine-tuned) | 33.5% | 26.8% | 57.9% | 46.8% |
| Δ (Fine-tuned - Base) | −3.1% | −4.3% | −4.5% | −5.8% |

Figure 16: Pass@1 performance of Llama3-8B (Base) and the Llama3-8B-Uni (fine-tuned) on other program synthesis benchmarks, including HumanEval, HumanEval+, MBPP, and MBPP+. Fine-tuning on the SIM dataset leads to a performance drop of $3 - 6\%$ on these program synthesis benchmarks.

distribution of the training data, leading to a temporary drop in performance. However, in later epochs after resampling (e.g., epoch 8), the performance of the resampling models outperforms that of the standard fine-tuning models, indicating the effectiveness of emulator-driven fine-tuning in improving fine-tuning performance.

### D.4 IMPACT OF DOMAIN-SPECIFIC FINE-TUNING ON OTHER BENCHMARKS

In Section 5.4, we have shown that fine-tuning on the domain dataset SIM leads to performance improvements on out-of-distribution tasks within the same domain, compared to the base model without fine-tuning. However, it remains uncertain whether fine-tuning on our domain dataset would also enhance performance on tasks from different domains, such as Python program synthesis tasks.

To investigate this, we evaluate our fine-tuned Llama3-8B-Uni model on other Python program synthesis benchmarks, including HumanEval (Chen et al., 2021), HumanEval+ (Liu et al., 2023b), MBPP (Austin et al., 2021), and MBPP+ (Liu et al., 2023b). Different from our benchmarks, these benchmarks focus on general Python program synthesis tasks from natural language or docstrings, without visual elements present in the benchmark tasks.

The results are presented in Figure 16. Our findings indicate that fine-tuning on our domain dataset SIM results in a slight performance drop ($3 - 6\%$) on these program synthesis benchmark tasks. We hypothesize that this is due to the SIM dataset's focus on visual programming tasks, which emphasize visual understanding, spatial reasoning, and planning—skills that are not directly applicable to other Python program synthesis tasks. Consequently, fine-tuning on our domain dataset does not provide additional knowledge for solving other benchmark tasks. Instead, the fine-tuning process may cause the model to forget some knowledge already acquired during the pre-training stage, leading to the performance drop in other benchmark tasks.

### D.5 CASE STUDY: OUTPUT CODE ANALYSIS ON PERTURBED TASKS

In the main paper, we presented a failure analysis by perturbing tasks and calculating the success rate. To illustrate the failure cases, we provide examples of output code from the evaluated models on these perturbed tasks, including GPT-4V, Llama3-70B, and Llama3-8B-Uni.

The output code is displayed in Figure 17. In the provided examples, we observe that GPT-4V has difficulty handling grid constraints and spatial reasoning. For example, in $T$ and $T_A$, GPT-4V attempts to reach the strawberry by ignoring the walls. However, once the walls are removed ($T_B$), GPT-4V is able to successfully solve the task. Interestingly, GPT-4V fails to solve $T_{A,B}$, even though this task is conceptually simpler than $T_B$ due to the absence of code constraints. Upon examining the code and the accompanying comments from GPT-4V, we found that it miscalculates the strawberry's coordinates, indicating a struggle with spatial reasoning. Additionally, we observed that moving the turtle closer to the strawberry consistently improves GPT-4V's performance, suggesting long-path planning and spatial reasoning are challenging for GPT-4V. However, for Llama3-70B and Llama3-8B-Uni, we observe that neither model successfully solves any of the perturbed tasks.

# E PROMPTS FOR THE XLOGOONLINE-MINI DOMAIN

In this section, we present three types of prompts for program synthesis in the XLogoOnline-Mini domain. Figures 18 and 19 show examples of the prompts using natural language and ASCII representation, respectively. Figure 20 shows the prompt for the few-shot + CoT prompting.

Note that after the title "#### Available Python Functions" in prompts, we provide an explanation and two examples of the code format. This is intended for *base models*, such as GPT-family and Llama-family base models, to ensure they follow the desired code format. However, *fine-tuning models* does not need this code format in the prompt, as models are trained with formatted code directly. Therefore, we omit the code format and examples from the prompts when fine-tuning models.

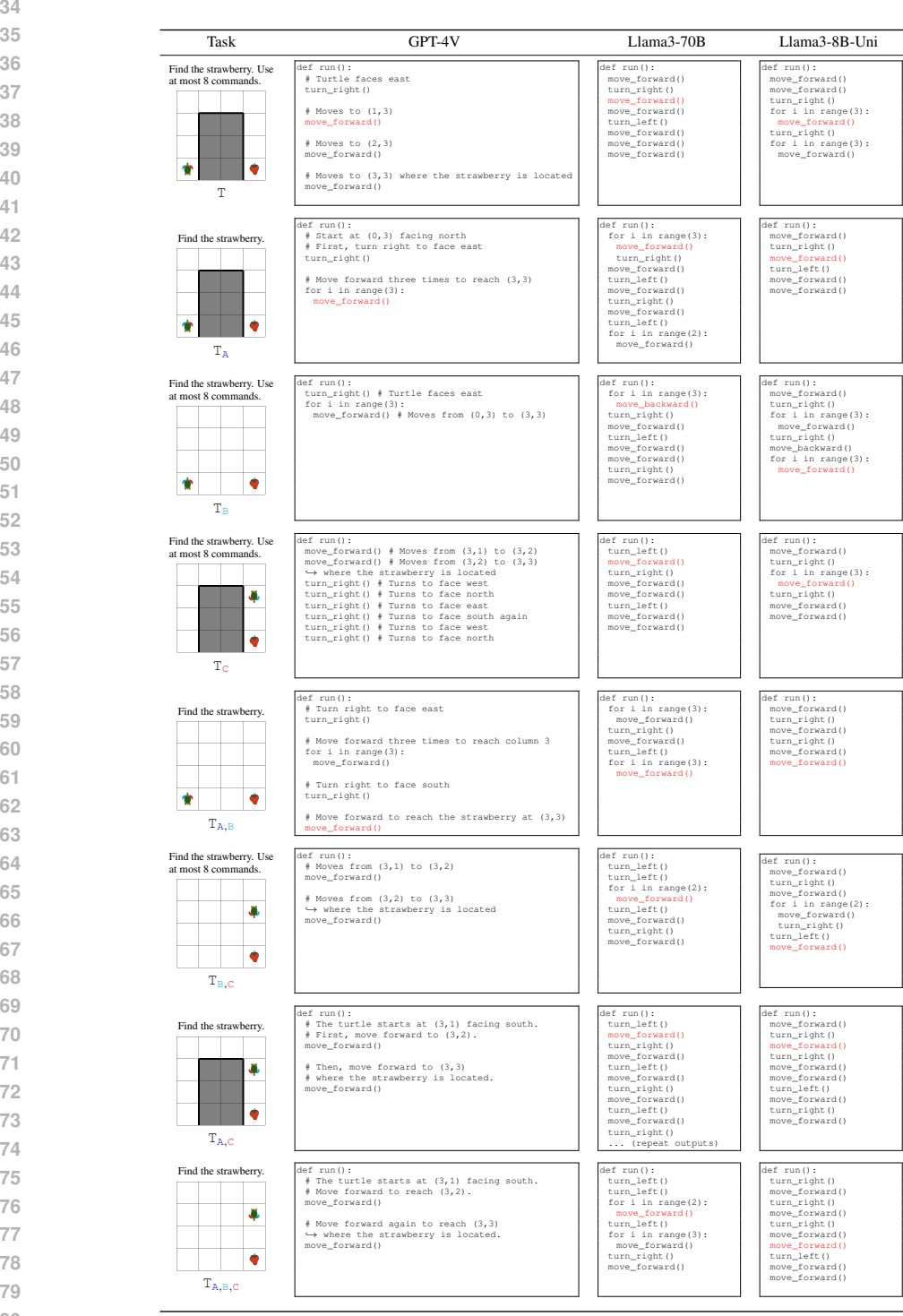

Figure 17: Output codes generated by GPT-4V, Llama3-70B, and Llama3-8B-Uni for various perturbations applied to a task T. The perturbations include removing code constraints ($T_A$), removing grid constraints ($T_B$), simplifying spatial relationships ($T_C$), and combinations of these perturbations ($T_{A,B}$, $T_{B,C}$, $T_{A,C}$, and $T_{A,B,C}$). Note that only the code is shown due to space limitations. The red line in the output code marks the point where the code first triggers an execution error or fails to successfully solve the task. GPT-4V successfully solves 5 out of 8 perturbed tasks, but Llama3-70B and fine-tuned Llama3-8B-Uni fail to solve any of the perturbed tasks.

## Natural Language Prompt for Code Generation in XLogoOnline-Mini

You are presented with a visual programming task involving a goal, a grid, a turtle, various items (or lines). You need to write Python code that enables the turtle to accomplish the goal within the grid.

#### Grid and Turtle
- The task has a `m x n` grid. The coordinates of the grid cells are `(x, y)`, where `x` is the column number and `y` is the row number. The top-left cell has coordinates `(0, 0)`. - The turtle starts at a specific grid cell and faces one of four directions: North, East, South, or West.

#### Items
Each item in the grid is defined by three attributes:
- `count`: The number of identical items in that grid cell.
- `color`: The item's color. Options include red, green, blue, yellow, black, white, orange, purple, and pink.
- `name`: The type of the item, such as circle, rectangle, triangle, cross, strawberry, or lemon.

#### Lines
Sometimes, the grid doesn't contain any items but has lines with colors. You need to draw lines of the specified color to solve the task.

#### Grid Cell Properties
- A grid cell may be `accessible` or `forbidden`. The turtle can move to an accessible cell but not into a forbidden cell. If the turtle tries to move into a forbidden cell, it will crash and fail to solve the task.
- Grid cells can have walls on their edges (top, bottom, left, and right). The turtle cannot move through walls, otherwise it will crash and fail to solve the task.

#### Available Python Functions
To solve the task, you can use the following Python functions:
- `move_forward()`: This function moves the turtle forward in the direction it is facing by one grid cell. For example, if the turtle is at the position (x, y) and facing north, after executing move_forward(), the turtle will be at the position (x, y-1).
- `move_backward()`: This function moves the turtle backward in the direction it is facing by one grid cell. For example, if the turtle is at the position (x, y) and facing west, after executing `move_backward()`, the turtle will be at the position (x+1, y).
- `turn_left()`: This function makes the turtle turn left in the direction it is facing - by 90 degrees. For example, if the turtle is facing north, after executing `turn_left()`, the turtle will be facing west.
- `turn_right()`: This function makes the turtle turn right in the direction it is facing - by 90 degrees. For example, if the turtle is facing south, after executing `turn_right()`, the turtle will be facing west.
- `setpc(color)`: This function sets the pen color to the specified color. The available colors are: red, green, blue, yellow, black, white. The default pen color is black. The trajectory of the turtle is drawn with the pen color.
- `for` loop: This loop is used to repeat a set of commands a specified number of times. For example, `for i in range(4):` will repeat the commands inside the loop 4 times.
Your code should follow the format:
```python
def run():
    # Your solution code goes here
    pass
```
Here are some examples of the code:
Example 1:
```python
def run():
    move_forward()
    for i in range(4):
        move_forward()
        turn_left()
```
Example 2:
```python
def run():
    move_forward()
    setpc('red')
    for i in range(3):
        move_forward()
    turn_right()
    move_backward()
```

Now, write a CORRECT Python code that successfully solves the following task.
### Task:
A 3x3 grid. The turtle starts at (1,1) facing north.
Accessible cells: (0,0), (1,0), (2,0), (0,1), (1,1), (2,1), (0,2), (1,2), (2,2).
Items in the grid:
- 1 red strawberry at (1,0).

### Goal:
Find the strawberry.

### CORRECT code:

Figure 18: An example of natural language prompt in the XLogoOnline-Mini domain.

# ASCII-based Prompt for Program Synthesis in XLogoOnline-Mini

You are presented with a visual programming task involving a goal, a grid, a turtle, various items (or lines). You need to write Python code that enables the turtle to accomplish the goal within the grid.

#### Grid and Turtle

A task's grid contain a turtle and some items. The turtle can face one of four directions: North (`^`), South (`v`), East (`>`), or West (`<`). An item has three attributes: `count`, `color`, and `name`. The `count` indicates the number of identical items in that grid cell. The `color` specifies the item's color, and the `name` describes the item's type. Here are the possible options:
- Colors: Red (`R`), Green (`G`), Blue (`B`), Yellow (`Y`), Black (`K`), White (`W`), Orange (`O`), Purple (`U`), Pink (`P`)
- Names: Circle (`o`), Rectangle (`□`), Triangle (`△`) ,Cross (`X`), Strawberry (`S`), Lemon (`L`)
- Counts: `1`, `2`, `3`, `4`
- For example, `2RS` means two red strawberries.

We use the following symbols to describe a grid:
- `—` represents the top or bottom edge of a grid cell.
- `|` represents the left or right edge of a grid cell.
- `===` represents an upper or lower wall of a cell.
- `||` represents a left or right wall of a cell.
- `+` represents the corner of a grid cell.
- `X` represents a forbidden cell that cannot be accessed.

#### Grid Cell Properties
- A grid cell may be `accessible` or `forbidden`. The turtle can move to an accessible cell but not into a forbidden cell. If the turtle tries to move into a forbidden cell, it will crash and fail to solve the task.
- Grid cells can have walls on their edges (top, bottom, left, and right). The turtle cannot move through walls, otherwise it will crash and fail to solve the task.

#### Available Python Functions
To solve the task, you can use the following Python functions:
- `move_forward()`: This function moves the turtle forward in the direction it is facing by one grid cell. For example, if the turtle is at the position (x, y) and facing north, after executing move_forward(), the turtle will be at the position (x, y-1).
- `move_backward()`: This function moves the turtle backward in the direction it is facing by one grid cell. For example, if the turtle is at the position (x, y) and facing west, after executing `move_backward()`, the turtle will be at the position (x+1, y).
- `turn_left()`: This function makes the turtle turn left in the direction it is facing - by 90 degrees. For example, if the turtle is facing north, after executing `turn_left()`, the turtle will be facing west.
- `turn_right()`: This function makes the turtle turn right in the direction it is facing - by 90 degrees. For example, if the turtle is facing south, after executing `turn_right()`, the turtle will be facing west.
- `setpc(color)`: This function sets the pen color to the specified color. The available colors are: red, green, blue, yellow, black, white. The default pen color is black. The trajectory of the turtle is drawn with the pen color.
- `for` loop: This loop is used to repeat a set of commands a specified number of times. For example, `for i in range(4):` will repeat the commands inside the loop 4 times.
Your code should follow the format:
```python
def run():
    # Your solution code goes here
    pass
```
Here are some examples of the code:
Example 1:
```python
def run():
    move_forward()
    for i in range(4):
        move_forward()
        turn_left()
```
Example 2:
```python
def run():
    move_forward()
    setpc('red')
    for i in range(3):
        move_forward()
    turn_right()
    move_backward()
```

Now, write a CORRECT Python code that successfully solves the following task:
### Task:
```
+---+---+---+
|   |1RS|   |
+---+---+---+
|   | ^ |   |
+---+---+---+
|   |   |   |
+---+---+---+
```
### Goal:
Find the strawberry.
### CORRECT Code:

Figure 19: An example of ASCII-based prompt in the XLogoOnline-Mini domain.

---

## **Few-shot + CoT Prompt for Code Generation in XLogoOnline-Mini**

You are presented with a visual programming task involving a goal, a grid, a turtle, various items (or lines). You need to write Python code that enables the turtle to accomplish the goal within the grid.

#### Grid and Turtle
- The task has a `m x n` grid. The coordinates of the grid cells are `(x, y)`, where `x` is the column number and `y` is the row number. The top-left cell has coordinates `(0, 0)`. - The turtle starts at a specific grid cell and faces one of four directions: North, East, South, or West.

#### Items
Each item in the grid is defined by three attributes:
- `count`: The number of identical items in that grid cell.
- `color`: The item's color. Options include red, green, blue, yellow, black, white, orange, purple, and pink.
- `name`: The type of the item, such as circle, rectangle, triangle, cross, strawberry, or lemon.

#### Lines
Sometimes, the grid doesn't contain any items but has lines with colors. You need to draw lines of the specified color to solve the task.

#### Grid Cell Properties
- A grid cell may be `accessible` or `forbidden`. The turtle can move to an accessible cell but not into a forbidden cell. If the turtle tries to move into a forbidden cell, it will crash and fail to solve the task.
- Grid cells can have walls on their edges (top, bottom, left, and right). The turtle cannot move through walls, otherwise it will crash and fail to solve the task.

#### Available Python Functions
To solve the task, you can use the following Python functions:
- `move_forward()`: This function moves the turtle forward in the direction it is facing by one grid cell. For example, if the turtle is at the position (x, y) and facing north, after executing move_forward(), the turtle will be at the position (x, y-1).
- `move_backward()`: This function moves the turtle backward in the direction it is facing by one grid cell. For example, if the turtle is at the position (x, y) and facing west, after executing `move_backward()`, the turtle will be at the position (x+1, y).
- `turn_left()`: This function makes the turtle turn left in the direction it is facing - by 90 degrees. For example, if the turtle is facing north, after executing `turn_left()`, the turtle will be facing west.
- `turn_right()`: This function makes the turtle turn right in the direction it is facing - by 90 degrees. For example, if the turtle is facing south, after executing `turn_right()`, the turtle will be facing west.
- `setpc(color)`: This function sets the pen color to the specified color. The available colors are: red, green, blue, yellow, black, white. The default pen color is black. The trajectory of the turtle is drawn with the pen color.
- `for` loop: This loop is used to repeat a set of commands a specified number of times. For example, `for i in range(4):` will repeat the commands inside the loop 4 times.

Your code should follow the format:
```python
def run():
    # Your solution code goes here
    pass
```

Here are some examples of the the tasks and their corresponding solution codes:
{few_shot_example_1}
{few_shot_example_2}
{few_shot_example_3}

Now, write a CORRECT Python code that successfully solves the following task.
### Task:
A 3x3 grid. The turtle starts at (1,1) facing north.
Accessible cells: (0,0), (1,0), (2,0), (0,1), (1,1), (2,1), (0,2), (1,2), (2,2).
Items in the grid:
- 1 red strawberry at (1,0).

### Goal:
Find the strawberry.

### CORRECT code:

Figure 20: An example of the few-shot + CoT prompt in the XLogoOnline-Mini domain. This prompt has several placeholders to include details for few-shot examples.

---

**{few_shot_example_1}**

### Task 1: A 4x4 grid. The turtle starts at (3,3) facing south.
Accessible cells: (3,0), (3,1), (3,2), (0,3), (1,3), (2,3), (3,3).
Forbidden cells: (0,0), (1,0), (2,0), (0,1), (1,1), (2,1), (0,2), (1,2), (2,2).
Lines in the grid:
- A red line from (3,0) to (3,1).
- A red line from (3,1) to (3,2).
- A red line from (3,2) to (3,3).
- A red line from (0,3) to (1,3).
- A red line from (1,3) to (2,3).
- A red line from (2,3) to (3,3).

### Goal 1:
Draw the picture in red. Use at most 8 commands.

### Code 1:
Explanation:
1. Starting Position and Initial Setup:
- The turtle starts at `(3, 3)`, facing south. To achieve the goal of drawing the picture in red, we first need to set the pen color to red using the `setpc('red')` command.

2. Plan the path:
- To draw the pattern, the path is `(3,3) -> (3,0) -> (3,3) -> (0,3)`.

3. Compiling the Path into Commands:
- The turtle starts at `(3, 3)`, facing south. To draw the first red line from `(3, 3)` to `(3, 0)`, we need to first set the pen color to red and then move back 3 times to reach from `(3, 3)` to `(3, 0)`.
- Now, the turtle is at `(3, 0)` and still facing south. To move back to `(3, 3)`, we move forward 3 times in the same direction (south).
- At `(3, 3)` and facing south, the turtle needs to turn right to face west.
- Now, the turtle is at `(3, 3)` and facing west. Move forward 3 times to reach from `(3, 3)` to `(0, 3)` facing west.
- Now the turtle has drawn the picture in red.

Putting it all together and notice that the solution code can use at most 8 commands. Here is the solution code:

```python
def run():
    setpc('red')
    for i in range(3):
        move_backward()
    for i in range(3):
        move_forward()
    turn_right()
    for i in range(3):
        move_forward()
```

Figure 21: The detail of the first few-shot + CoT example {few_shot_example_1}.

{few_shot_example_2}

### Task 2:
A 3x3 grid. The turtle starts at (1,2) facing south.

Accessible cells: (0,0), (1,0), (2,0), (0,1), (1,1), (2,1), (0,2), (1,2), (2,2).
(1,0) has a wall at the bottom edge.
(2,0) has a wall at the bottom edge.
(1,1) has a wall at the top edge.
(2,1) has a wall at the top edge.

Items in the grid:
- 4 red strawberries at (1,0).
- 1 red strawberry at (2,0).
- 2 red strawberries at (2,1).
- 1 red strawberry at (2,2).

### Goal 2:
Collect exactly 5 strawberries.

### Code 2:
Explanation:
1. Starting Position and Initial Setup:
- The turtle starts at `(1, 2)`, facing south.
- The goal is to collect exactly 5 strawberries.
- There are 4 strawberries at `(1, 0)` and 1 strawberry at `(2, 0)`. By visiting these two cells, we can collect exactly 5 strawberries. We need a path to visit both `(1, 0)` and `(2, 0)` while considering walls that block movement between cells `(1, 0)` & `(1, 1)` and `(2, 0)` & `(2, 1)`.

2. Planning the Path:
- An ideal path to avoid the walls and collect strawberries is: `(1, 2) -> (1, 1) -> (0, 1) -> (0, 0) -> (2, 0)`.
- This path efficiently collects exactly 5 strawberries.

3. Compiling the Path into Commands:
- The turtle starts at `(1, 2)` facing south.
- To move from `(1, 2)` to `(1, 1)`, it needs to move backward. Now the turtle is at `(1, 1)` facing south.
- To move from `(1, 1)` to `(0, 1)`, it needs to turn right and move forward. Now it is at `(0, 1)` facing west.
- To move from `(0, 1)` to `(0, 0)`, it needs to turn right to face north, then move forward. Now it is at `(0, 0)` facing north.
- To move from `(0, 0)` to `(2, 0)`, it needs to turn right to face east, and finally, move forward twice to go from `(0, 0)` to `(2, 0)`.
- Now the turtle has collected exactly 5 strawberries.

Putting it all together. The solution code is:

```python
def run():
    move_backward()
    turn_right()
    move_forward()
    turn_right()
    move_forward()
    turn_right()
    move_forward()
    move_forward()
```

Figure 22: The detail of the second few-shot + CoT example {few_shot_example_2}.

1458
1459
1460
1461
1462
1463
1464
1465
1466
1467
1468
1469
1470
1471
1472
1473
1474
1475
1476
1477
1478
1479
1480
1481
1482
1483
1484
1485
1486
1487
1488
1489
1490
1491
1492
1493
1494
1495
1496
1497
1498
1499
1500
1501
1502
1503
1504
1505
1506
1507
1508
1509
1510
1511

**{few_shot_example_3}**

### Task 3:
A 3x3 grid. The turtle starts at (2,2) facing south.
Accessible cells: (0,0), (1,0), (2,0), (0,1), (1,1), (2,1), (0,2), (1,2), (2,2).
Items in the grid:
- 1 blue triangle at (0,0).
- 1 red circle at (1,0).
- 1 green circle at (2,0).
- 1 blue circle at (0,1).
- 1 blue rectangle at (1,1).
- 1 red triangle at (2,1).
- 1 red cross at (0,2).
- 1 green rectangle at (1,2).

### Goal 3:
Find the blue triangle without standing on a red. Use at most 3 'backward'.

### Code 3:
Explanation:
1. Starting Position and Initial Setup:
- The turtle starts at `(2, 2)`, facing south.
- The goal is to find the blue triangle located at `(0, 0)` without standing on any red items.
- There are several items in the grid: red circle at `(1, 0)`, red triangle at `(2, 1)`, and red cross at `(0, 2)`.
- We need to avoid these red items and reach `(0, 0)` using at most 3 backward movements.

2. Planning the Path:
- Since we have to avoid red items and should use at most 3 backward movements, we need to plan an efficient path.
- A suitable path that avoids red items and stays within the permitted movements is: `(2, 2) -> (1, 2) -> (1, 1) -> (0, 1) -> (0, 0)`.

3. Compiling the Path into Commands:
- The turtle starts at `(2, 2)` facing south. First, turn left to face east.
- Move backward to reach `(1, 2)` facing east.
- Turn left again to face north.
- Move forward to reach `(1, 1)` facing north.
- Turn right to face east.
- Move backward to reach `(0, 1)` facing east.
- Turn right to face south.
- Move backward to reach `(0, 0)` facing south.

Putting it all together. The solution code is:

```python
def run():
turn_left()
move_backward()
turn_left()
move_forward()
turn_right()
move_backward()
turn_right()
move_backward()
```

Figure 23: The detail of the third few-shot + CoT example {few_shot_example_3}.

