# OpenReview forum: "Program Synthesis Benchmark for Visual Programming in XLogoOnline Environment"
_ICLR.cc/2025/Conference — Submitted to ICLR 2025_

### Official Review · Reviewer_KKKK · 2024-11-02

**Soundness:** 3
**Presentation:** 4
**Contribution:** 3
**Rating:** 8
**Confidence:** 4

**Summary:**

This paper introduces a new visual reasoning benchmark that requires a combination of skills to be solved (like logic, math, coding, visual understanding). Then, they benchmark current visual language models, and show that they do not perform well on this benchmark. Finally, they propose an approach to improve such models, by finetuning them on more data that follows the distribution of the benchmark, as well as by doing it following a curriculum.

**Strengths:**

1 - This paper contributes a benchmark to study the capabilities of VLMs on program synthesis for visual tasks. The benchmark is challenging enough for current SOTA models to fail at it, but at the same time it is easy enough for humans to solve, making it a good contribution for the community. It also requires a variety of skills that previous benchmarks do not need, making it more general.

2 - The contributions are well motivated and well presented, making it an easy paper to read.

3 - The fine-tuning strategy using curriculum learning based on the harder and easier task is very sensible in this setting where multiple skills need to be learned. The specific implementation is reasonable, and provides very good results.

4 - The analysis (ablation study based on difficulty of the tasks, as well as failure analysis) is informative both in terms of understanding the benchmark better, but also in terms of understanding the models better.  Also, the "transferable skills" section is very pertinent. I was actually thinking about the zero-shot capabilities to unseen tasks before I got to that section.

**Weaknesses:**

I would appreciate if the authors analyzed the following:

1 - What is the drop (or increase) in performance in other standard vision tasks (such as VQA, classification) or even language or coding tasks when fine-tuning on the SIM dataset? It seems like the tasks follow a very specific pattern that may not be very related to other tasks. The paper shows that training on the SIM dataset contributes to tasks that were not part of the dataset, but I wonder if there is any contribution (positive or negative) to other tasks that follow different distributions (even though they may require similar visual or reasoning skills)

2 - What is the role of the prompt engineering in solving these tasks? How much effort and what techniques were used to ensure that the input prompts are not a limitation for the different models that were tested?

**Questions:**

- I would probably not call the REAL dataset "real-world" tasks. They are pretty much very synthetic and toy tasks, not real-world. Maybe you can refer to them as "existing" (or any other word). The term "real-world" was confusing at the beginning.  It would also help if you shoed an example of a SIM task in Figure 1, or in Section 4.

---

> ### Author Response · Authors · 2024-11-20
> **Response to Reviewer KKKK**
>
> Dear Reviewer,
>
> We appreciate your time and great feedback for improving our work! We would like to address each of your concerns below.
>
> ---
>
> ### Comment 1: "What is the drop (or increase) in performance in other tasks..."
>
> This is a very interesting experiment! To answer this question, we have conducted experiments on the HumanEval [1], HumanEval+ [2], MBPP [3], and MBPP+ [4] tasks. These are popular program synthesis tasks. The results are shown below:
>
>
> | Model                    | HumanEval | HumanEval+ | MBPP   | MBPP+  |
> |-------------------------|-----------|------------|--------|--------|
> | Llama3-8B (Base)  | 36.6%     | 31.1%      | 62.4%  | 52.6%  |
> | Llama3-8B-Uni (Fine-tuned)   | 33.5%     | 26.8%      | 57.9%  | 46.8%  |
> | Δ (Fine-tuned - Base)         | -3.1%     | -4.3%      | -4.5%  | -5.8%  |
>
>
> We found that fine-tuning on the SIM dataset leads to a small performance drop (3-6%) on other program synthesis benchmark tasks. Similar phenomena have also been observed in other works such as [5,6]. We hypothesize this is because the SIM dataset focuses on visual programming tasks which place more focus on visual understanding, spatial reasoning, and planning, which are not directly related to other Python program synthesis tasks. Therefore, it does not bring additional knowledge to solving other benchmark tasks in HumanEval and MBPP. Instead, such a fine-tuning process may cause the model to forget some knowledge that it has learned from the previous training stage, finally resulting in the performance drop in other benchmark tasks. I hope this could provide some insights into your question.
>
> We have also included this interesting experiments in the updated version of our paper (See Appendix D.4).
>
> ---
>
> ### Comment 2: "What is the role of the prompt engineering in solving these tasks?..."
>
> We appreciate the reviewer's feedback and would like to elaborate on our efforts in prompt engineering.
>
> We developed the prompt in the following manner:
>
> First, we considered using natural language to describe the visual task grid, as it is the most natural and straightforward way for LLMs to understand. When creating the natural language prompts, we began by drafting an initial version and then requesting outputs from GPT-4 and Llama3-70B. We analyzed these models' outputs and adjusted the prompt to ensure that if the output code was incorrect, it wasn’t due to missing information in the prompt. If any information was missing, we incorporated it into the prompt. We repeated this process several times for multiple tasks in our evaluation dataset. Eventually, we arrived at a final version of the prompt and then used ChatGPT to refine it further to ensure clarity and eliminate ambiguity.
>
> However, since our domain includes the visual task grid, natural language may not be the optimal representation. Therefore, we also investigated whether we could represent the visual task grid differently. Specifically, we examined an ASCII representation of the task grid, as illustrated in Figure 14 (See Appendix). Through our experiments, we found that natural language descriptions were more effective than ASCII representations for both GPT-4 and Llama3-70B models. However, when fine-tuning the models using these representations, the differences in effectiveness between ASCII-based and natural language-based prompts were minimal.
>
> Due to our visual programming nature, we also tried to use the visual task image as an additional input. The prompt was almost identical, except we explicitly stated that the image is also provided.
>
> These are the efforts we made in prompt engineering for our domain. I hope this clarifies your concerns.
>
> ---
>
> ### Comment 3: "I would probably not call the REAL dataset "real-world" tasks..."
>
> Thanks for this great feedback! We agree that the term "real-world" might be misleading. However, updating the term in the paper may confuse other reviewers at current stage. So we keep the term as it is in the current version. However, we promise to use a more appropriate term in the final version after the rebuttal phase. Thank for this great point!
>
> ---
> ### Comment 4: "It would also help if you shoed an example of a SIM task"
>
> Thank you for this suggestion! We've added a few examples of a SIM task in the Appendix (See Figure 12) due to the page limit. We hope this can provide a better understanding of the tasks in the SIM dataset.
>
>
> **References:**
>
> [1] https://github.com/openai/human-eval
>
> [2] https://huggingface.co/datasets/evalplus/humanevalplus
>
> [3] https://huggingface.co/datasets/google-research-datasets/mbpp
>
> [4] https://huggingface.co/datasets/evalplus/mbppplus
>
> [5] Towards Reliable Latent Knowledge Estimation in LLMs: In-Context Learning vs. Prompting Based Factual Knowledge Extraction. https://arxiv.org/pdf/2404.12957
>
> [6] Fine-Tuning or Fine-Failing? Debunking Performance Myths in Large Language Models. https://arxiv.org/abs/2406.11201

---

> > ### Comment · Reviewer_KKKK · 2024-11-25
> > **Keep the rating**
> >
> > I appreciate the response by the authors.
> >
> > The drop in performance in other coding tasks is, in my opinion, important to mention. However, I believe the paper's contributions still hold. Identifying problems where current VLMs do not perform well, and obtaining data for them, is the approach that has been shown to work in the past years.
> >
> > I keep the rating.

---

> > > ### Author Response · Authors · 2024-11-26
> > >
> > > Dear Reviewer,
> > >
> > > Thank you for your valuable feedback. We have updated our manuscript to include the performance drop as a footnote in the main paper, with more detailed results included in the Appendix. We truly appreciate your recognition of the paper's contributions!

---

### Official Review · Reviewer_nRED · 2024-11-04

**Soundness:** 4
**Presentation:** 4
**Contribution:** 3
**Rating:** 8
**Confidence:** 4

**Summary:**

This paper proposes XLOGOMINIPROG, a program synthesis benchmark in visual programming that requires a combination of different skills. They evaluate the performance of various large models (GPT, Llama, LLaVA) which show a surprisingly low success rate.
Additionally, they finetune a Llama3-8B model on a large-scale synthetic dataset of 80K visual programming tasks, outperforming GPT-4V and LLama3-70B. Finally, they propose emulator-driven fine-tuning, which designs curriculum over training data distribution by leveraging emulator feedback, i.e., assigning higher weights to tasks where the model struggles. The paper includes an analysis of failure cases, capabilities, and transfer learning skills.

**Strengths:**

* This paper introduces a new program synthesis benchmark for the *visual programming domain*, which currently lacks benchmarking compared to program synthesis benchmarks from natural language or docstrings (e.g., HumanEval, MBPP, APPS).
* Experiment results are comprehensive across diverse model types and scale, along with an in-depth analysis of various failure types and capabilities. It is interesting to see that current models dominantly lack spatial reasoning, especially GPT-4V.
* Emulator-driven fine-tuning could be extended to other program synthesis domains that contain a wide range of complexity in the training set.
* The paper is well-organized and written clearly with sufficient detail on the dataset statistics, generation process, and experimental setup.

**Weaknesses:**

* The paper proposes a new benchmark focusing on *visual programming*, yet only provides results on two VLM families: GPT-4V and LLaVA-1.5. As there exists a plethora of open-sourced VLMs, I think it would only make the paper better by adding diverse state-of-the-art models (e.g., Llama-3.2-vision, LLaVA-onevision, Phi-3-Vision, Phi-3.5-Vision, QwenVL, Cameleon, LLaVA-next, Pixtral).
* Need clarity in the data generation process. Are there cases where the generated code is correct for the wrong reasons (e.g., logic incorrect, answer correct)? Is the emulator able to filter out these cases during the correctness check? If not, how does that affect the performance of fine-tuned models trained on the synthetic dataset?
* Lacks detailed explanation of the emulator.

**Questions:**

* How do the benchmark results look like on other open-sourced VLMs?
* It's very interesting that fine-tuning a VLM (Llava1.5-13B-Uni) performs worse than fine-tuning an LLM (Llama2, Llama3), while the vision model (GPT-4V) outperforms GPT-4 in the base models experiment. Why do you think this is the case? It would be interesting to compare fine-tuning a more state-of-the-art VLM (e.g., Llama-3.2-vision, LLaVA-onevision).
* How does the emulator validate the soundness of the generated code? For instance, some code may be correct for the wrong reasons (e.g., logic incorrect, answer correct). Is the emulator properly filtering out these cases?

---

> ### Author Response · Authors · 2024-11-20
> **Response to Reviewer nRED**
>
> Dear Reviewer,
>
> We appreciate your feedback and would like to address each of your concerns.
>
> ---
>
> ### Comment 1: "The paper only provides results on two VLM families..."
>
> Thanks for this valuable feedback. We've added 7 new multimodal models in our evaluation. Here is a summary of the success rates for all the multimodal models evaluated in our paper:
>
> | Model | REAL (%) | SIM-EVAL (%) |
> |-------|----------|--------------|
> | GPT-4V | 20.00 | 5.50 |
> | Llava1.5-7B | 0.00 | 0.00 |
> | Llava1.5-13B | 0.00 | 0.00 |
> | InternVL2-8B | 0.00 | 0.00 |
> | InternVL2-Llama3-76B | 9.41 | 1.50 |
> | Qwen2VL-7B | 0.00 | 0.20 |
> | Qwen2VL-72B | 0.00 | 0.40 |
> | NVLM-D-72B | 1.18 | 2.00 |
> | Molmo-7B-D | 0.00 | 0.60 |
> | Molmo-72B | 1.18 | 0.40 |
>
> We have also updated our paper to reflect these changes (See Table 6).
>
> ---
>
> ### Comment 2: "Need clarity in the data generation process..."
>
> This is an great question! In our case, all the generated (task, code) pairs in our dataset are correct. Since our method does not involve LLMs and does not generate explanations, reasoning, or comments for the code, there are no cases where the code is correct for the wrong reasons. Additionally, we have included a more detailed explanation of our dataset generation process in Appendix B.1. We hope this clarifies your concerns!
>
> ---
> ### Comment 3: "Lacks detailed explanation of the emulator..."
>
> Thanks for the feedback. Our emulator runs the code for a given task and providing detailed execution results and messages, similar to a Python interpreter. This process is designed to be precise and unambiguous.
>
> The emulator operates in the following way:
> 1. Given a (task, code) pair in our domain, the emulator runs the code for the task and then returns the execution result.
> 2. During execution, the emulator checks the code format, execution crashes, code constraints, and whether the code meets the task's goal. The code constraints and the task's goal are specified in JSON format for precise and unambiguous checking. When creating prompts for models to generate the code, these code constraints and goals were translated into natural language using a fixed translation template. For example, a task's code constraints and goal might be translated as "Find the strawberry using at most 8 commands."
> 3. After all above checks are performed, the emulator finally provides the execution result, which is either `success` or an error message indicating the specific reason for the failure. For example, when code execution is successful for a task, the execution result is `success`. If there is an error, such as `hitting the wall`, the emulator generates the appropriate error type and message.
>
> We have noticed that this might be not clear in the original paper. So we've incorporated this clarification in the updated version of the paper in the Appendix (see Section C). Please let us know if you have any further suggestions or concerns. We're happy to address and improve the work.
>
> ---
>
> ### Comment 4: "It's very interesting that fine-tuning a VLM (Llava1.5-13B-Uni) performs worse than fine-tuning an LLM (Llama2, Llama3)..."
>
> Thank you for your insightful question. We have also been thinking about this question. We list some of our thoughts for potential reasons below:
>
> First, the choice of base model may affect performance. Llava1.5-13B is built on the Llama1 model, which may limit its performance when compared to Llama2 and Llama3. We've noticed that fine-tuning more advanced models typically results in better performance (for example, fine-tuning Llama3 yields better performance compared to fine-tuning Llama2). Therefore, fine-tuning a state-of-the-art vision language model with an improved base model (e.g., Llama3.1, Qwen2) might also enhance its performance.
>
>
> Besides, we observed significant instability in Llava1.5-13B-Uni's performance during fine-tuning. We found that, after fine-tuning Llava1.5-13B with 5 different seeds, only one seed achieved performance comparable to the fine-tuned Llama3 models (around a 54% success rate). The other 4 seeds failed to reach similar levels of performance, resulting in an almost zero success rate. This is also why the standerr for Llava1.5-13B is much higher than fine-tuning other models. This may suggest that a more complex model with both visual and language components might require more careful fine-tuning to achieve stable performance. In our experiments, the visual component of the Llava model is not being fine-tuned. If the visual encoder has inherent limitations in understanding the task images, the integrated image may also confuse the model instead of enhancing its performance.
>
> These are our thoughts about the potential reasons. There may be other factors at play that we have not considered. We will keep exploring this question in the future. Thanks for this insightful question! We appreciate your feedback and are happy to address any further concerns.

---

> ### Author Response · Authors · 2024-11-30
> **A gentle reminder**
>
> Dear Reviewer nRED,
>
> Thank you for recognizing our work! We hope our responses have addressed your concerns and questions. As the discussion period is coming to a close, we would greatly appreciate any further input you may have. Thanks again for your great feedback!

---

> > ### Comment · Reviewer_nRED · 2024-12-03
> >
> > Thank you for your detailed response. The rebuttal and updated PDF have addressed all of my concerns. In particular, the response to Comment 1 demonstrates that the authors provide a visual programming benchmark that is challenging to solve for very large, state-of-the-art VLMs. I believe the paper makes a solid contribution, and I am maintaining my score of 8: accept, good paper.

---

> > > ### Author Response · Authors · 2024-12-04
> > > **Thank you**
> > >
> > > We are pleased to have addressed all your concerns. Thank you for recognizing our work and helping us improve the paper!

---

### Official Review · Reviewer_81e9 · 2024-11-04

**Soundness:** 2
**Presentation:** 2
**Contribution:** 2
**Rating:** 3
**Confidence:** 5

**Summary:**

The paper presents a novel benchmark for evaluating the performance of large language and multimodal models on tasks that require a combination of skills such as spatial planning, basic programming, and logical reasoning within the XLogoOnline visual programming environment.

**Strengths:**

1. Novel Programming Generation Evaluation: The evaluation of programming generation based on the XLogoOnline visual programming environment is somewhat novel, as it integrates vision, language, and coding skills to assess LLMs.

2. Well-Structured Paper Organization: The paper is well-organized, starting with an introduction to the benchmark, followed by a detailed presentation of the training data, and concluding with an evaluation of different performance metrics and analyses.

**Weaknesses:**

1. Lack of Task Challenge: The task presented in this paper appears to lack sufficient challenge, as evidenced by Table 6. Although the zero-shot performance of both closed and open-source models is low, simply synthesizing a large amount of training data can significantly boost performance. This suggests that the task is not particularly challenging. The poor performance of current models can be attributed to their lack of training with this specific data. A more detailed analysis of task difficulty is needed to solve this issue.

2. Narrow Scope of Visual Programming: The use of the XLogoOnline environment limits the code action space, as illustrated in Figure 1, which further contributes to the task's lack of challenge. Additionally, other visual programming languages, such as Scratch, are not considered in this work. This narrow focus limits the domain of the study, making it insufficient for a top-tier conference like ICLR.

3. Insufficient Evaluation of Large Multimodal Models: The paper evaluates only three multimodal models, leaving unclear whether current state-of-the-art closed and open-source multimodal models, such as Claude3.5, Gemini, and Qwen2-VL, could achieve better performance. A more comprehensive evaluation with a broader range of models is necessary to draw meaningful conclusions.

**Questions:**

Check Weaknesses.

---

> ### Author Response · Authors · 2024-11-20
> **Response to Reviewer 81e9**
>
> Dear Reviewer,
>
> Thanks for your constructive feedback! We appreciate your feedback and would like to address each of your concerns.
>
> ---
>
> ### Comment 1: "Lack of Task Challenge..."
>
> > Lack of Task Challenge: The task presented in this paper appears to lack sufficient challenge, as evidenced by Table 6...
>
> To understand whether our benchmark is challenging, we would like to compare with other related benchmarks.
>
> For example, we can compare our benchmark with MMMU [1,2], a widely used multimodal benchmark:
>
> 1. On MMMU (Val), GPT-4V achieves 56% accuracy without fine-tuning (https://mmmu-benchmark.github.io/#leaderboard).
> 2. On our benchmark, GPT-4V only achieves 20% accuracy without fine-tuning (See Table 6 in our paper).
> 3. Even our best fine-tuned model (Llama3-8B-Emu) achieves 60.23% accuracy, which is similar to GPT-4V's performance on MMMU. There is still a large room for further improvement on our benchmark.
>
> We can also compare our benchmark with other popular program synthesis benchmarks, such as MBPP [3] and HumanEval [4]:
>
> 1. On MBPP [3], GPT-4 achieves 73.3% pass@1 accuracy. On HumanEval [4], GPT-4 achieves 86.6% pass@1 accuracy.
> 2. On our benchmark, GPT-4 achieves 12.94% accuracy (See Table 6 in our paper).
> 3. The performance of GPT-4 on our benchmark is much lower than its performance on MBPP and HumanEval, indicating that our benchmark is inherently challenging. We believe this is because our benchmark requires multiple combined skills, such as spatial reasoning, programming skills, etc.
>
>
> Finally, we believe that the challenges of our benchmark come from the fact that the tasks are originally designed for _K-2 students_, making them quite simple for human experts. In comparison, tasks in other benchmarks (e.g., MBPP, HumanEval) are designed for beginners to entry-level programmers, which are much harder than our tasks. However, we observed that models' performance on our benchmark is still far behind their performance on MBPP and HumanEval, indicating that our benchmark is quite challenging and interesting for further study. We had also provided more insights to understand why models fail on our benchmark (See Section 5.3).
>
>
> ---
>
> ### Comment 2: "Narrow Scope of Visual Programming..."
>
>
> Thank you for the feedback! Regarding your concerns about the narrow scope of visual programming and the limited code space, we would like to clarify from the perspectives of both task space and code space.
>
> First, while our benchmark focuses on the XLogoOnline environment, it maintains a broad task space. As illustrated in Figure 1, our tasks span diverse types of visual reasoning, each requiring different programming concepts and spatial understanding (e.g., math, geometry, etc.). The combination of varied task types and different code constraints creates a large task space.
>
> Second, the simplified code space in our benchmark also creates unique challenges. While this simplified code space makes tasks easier for humans, it actually poses significant difficulties for models. Our failure analysis (See Section 5.3) shows that models often struggle to adhere to the constrained code space, frequently generating invalid commands from other domains (e.g., attempting to use `turn_around()`, which is not part of our command set). This indicates that models face challenges when operating within a more restricted code space.
>
>
> ---
>
> ### Comment 3: "Insufficient Evaluation of Large Multimodal Models..."
>
>
> Thanks for this valuable feedback! According to your suggestion, we have expanded our model evaluation since the initial submission. We've added 7 new multimodal models in our evaluation. Here is a summary of the success rates for all the multimodal models evaluated in our paper:
>
> | Model | REAL (%) | SIM-EVAL (%) |
> |-------|----------|--------------|
> | GPT-4V | 20.00 | 5.50 |
> | Llava1.5-7B | 0.00 | 0.00 |
> | Llava1.5-13B | 0.00 | 0.00 |
> | InternVL2-8B | 0.00 | 0.00 |
> | InternVL2-Llama3-76B | 9.41 | 1.50 |
> | Qwen2VL-7B | 0.00 | 0.20 |
> | Qwen2VL-72B | 0.00 | 0.40 |
> | NVLM-D-72B | 1.18 | 2.00 |
> | Molmo-7B-D | 0.00 | 0.60 |
> | Molmo-72B | 1.18 | 0.40 |
>
> We have also updated our paper to reflect these changes (See Table 6).
>
> ---
>
> Thank you for your valuable feedback for improving our work. We hope our responses have addressed your concerns and you would consider raising the score. If you have any further questions, please feel free to reach out and we are more than happy to answer them!
>
>
>
> **References:**
>
> [1] https://evalplus.github.io/leaderboard.html
>
> [2] MMMU: A Massive Multi-discipline Multimodal Understanding and Reasoning Benchmark for Expert AGI. https://arxiv.org/abs/2311.16502.
>
> [3] MBPP: Program Synthesis with Large Language Models. https://arxiv.org/abs/2108.07732.
>
> [4] HumanEval: Evaluating Large Language Models Trained on Code. https://arxiv.org/abs/2107.03374.

---

> > ### Comment · Reviewer_81e9 · 2024-12-03
> > **Response**
> >
> > Thank you for your responses. However, they do not fully address my concerns.
> >
> > **Regarding the first point:** Comparing your work to the MMMU benchmark does not sufficiently address the issue of limited challenge. Fine-tuning models like LLaMA 3 with a domain-specific dataset can boost their success rate to around 60%. This result suggests that the low performance of current models is primarily due to the task being out-of-domain rather than inherently challenging. To improve performance, similar data could simply be included in their training recipes. Therefore, the main contribution of this work seems to be a reminder that current models underperform in this domain and require additional domain-specific data. Once such data is included, performance improves significantly, indicating that this task is not particularly challenging.
> >
> > **Regarding the second point:** The response does not adequately address the concern about the narrow domain. Focusing solely on the XLogoOnline environment limits the evaluation to the Logo programming language. To achieve high performance under these constraints, models need to optimize specifically for Logo, potentially compromising their generalization to other languages, such as Scratch. This specialization undermines their broader applicability, and the authors' response does not resolve this issue.
> >
> > For these reasons, I keep my original rating.

---

> > > ### Author Response · Authors · 2024-12-03
> > > **Response to Reviewer 81e9**
> > >
> > > Dear Reviewer,
> > >
> > > Thanks for your follow-up feedback! We would like to address your concerns one by one.
> > >
> > > ---
> > >
> > > > **Regarding the first point**: Comparing your work to the MMMU benchmark does not sufficiently address the issue of limited challenge. Fine-tuning models like LLaMA 3 with a domain-specific dataset can boost their success rate to around 60%. This result suggests that the low performance of current models is primarily due to the task being out-of-domain rather than inherently challenging. To improve performance, similar data could simply be included in their training recipes. Therefore, the main contribution of this work seems to be a reminder that current models underperform in this domain and require additional domain-specific data. Once such data is included, performance improves significantly, indicating that this task is not particularly challenging.
> > >
> > > We believe that **most existing benchmarks could benefit from fine-tuning on domain-specific data—not just our benchmark.** However, this does not imply that these benchmarks lack challenges.
> > >
> > > Instead, we argue that most existing benchmarks, including ours, focus more on evaluating the **emergent capabilities** of large foundation models. This is because large foundation models have shown surprising emergent capabilities that were neither specifically trained for nor anticipated. Our paper introduces a benchmark to assess how well these large foundation models can solve problems in the visual programming domain, particularly **whether they exhibit strong emergent capabilities to solve visual programming tasks they have never seen during training.** From this perspective, our benchmark is particularly challenging and highlights that large foundation models still fall short in spatial reasoning.
> > >
> > >
> > > ---
> > >
> > > > **Regarding the second point:** The response does not adequately address the concern about the narrow domain. Focusing solely on the XLogoOnline environment limits the evaluation to the Logo programming language. To achieve high performance under these constraints, models need to optimize specifically for Logo, potentially compromising their generalization to other languages, such as Scratch. This specialization undermines their broader applicability, and the authors' response does not resolve this issue.
> > >
> > >
> > >
> > > We would like to address this from two perspectives:
> > >
> > > 1. From the perspective of evaluating large foundation models (without fine-tuning):
> > > 	- We believe that most existing program synthesis benchmarks focus on a single programming language (e.g., Python). Our benchmark uses the Python-like Logo programming language. We would like to clarify that the Logo programming language is not the most interesting or challenging aspect. Instead, **the real challenge lies in the complex spatial elements of the visual tasks and how to combine different skills to solve these tasks**, which makes them particularly challenging and not considered in other program synthesis benchmarks. When evaluating large foundation models without fine-tuning, these models do not need to be specifically optimized for the Logo programming language.
> > >
> > > 2. From the perspective of evaluating fine-tuned models:
> > > 	- We admit that **fine-tuned models** need to be optimized specifically for Logo programming language and may not generalize well to Scratch programming tasks. However, this is also a well-known issue across other benchmarks. For instance, fine-tuning models on Python programming tasks may not generalize well to tasks in other domain-specific programming languages.
> > >
> > >
> > > We agree that incorporating other visual programming languages could be an interesting direction for future research. However, the main focus of our benchmark is to **evaluate and understand how well large models can combine different skills**. We believe that the various task types and constraints already provide a comprehensive evaluation of this aspect. However, we also appreciate your suggestion.
> > >
> > > ---
> > >
> > > We hope our responses have addressed most of your concerns and you would consider raising the score. Thanks again for your valuable feedback!

---

> ### Author Response · Authors · 2024-11-25
> **A gentle reminder**
>
> We hope that our responses have addressed the reviewer's concerns and questions. Given that the discussion period is ending soon, we would greatly appreciate the reviewer's input. If you have more questions, we would be glad to provide further responses. Thank you for your feedback!

---

> > ### Author Response · Authors · 2024-11-30
> > **Follow-up reminder**
> >
> > Dear Reviewer 81e9,
> >
> > We hope our previous responses have addressed your concerns. As the rebuttal period is coming to a close, we would greatly appreciate any additional feedback or input you may have. If there are any remaining questions or points to clarify, we would be happy to provide further responses. Thank you again for your time and consideration!

---

### Official Review · Reviewer_znzb · 2024-11-07

**Soundness:** 2
**Presentation:** 3
**Contribution:** 2
**Rating:** 5
**Confidence:** 3

**Summary:**

The paper introduces a new benchmark, which is designed to evaluate the performance of large language models on program synthesis tasks within the XLogoOnline visual programming environment. The benchmark includes real-world and synthetic tasks that require a combination of skills such as spatial planning, basic programming, and logical reasoning. The study found that current state-of-the-art models like GPT-4V and Llama3-70B struggle with these tasks, achieving low success rates. To improve model performance, the authors developed a fine-tuning pipeline using a large-scale synthetic training dataset, which significantly boosted the performance of the Llama3-8B model. Additionally, they showcased how emulator-driven feedback can be used to design a curriculum over training data distribution, further enhancing the performance of fine-tuned models. The paper provides an in-depth failure analysis to understand model limitations and will publicly release the benchmark for future research on program synthesis in visual programming.

**Strengths:**

It introduces XLOGOMINIPROG, a new benchmark that tests multiskill tasks in visual programming, an area where current models perform poorly.
It demonstrates the effectiveness of emulator-driven feedback for designing a dynamic training curriculum, further boosting model performance.
Detailed experiments were carried out

**Weaknesses:**

The emulator-driven fine-tuning provides only binary correctness feedback on the predicted code, which might not be sufficient for identifying and correcting specific errors in the generated code. Can it be in natural language?
There's a risk that models could overfit to the synthetic data used in training, which might not perfectly mimic the complexity and variability of real-world programming challenges. I think this is the most serious disadvantage

**Questions:**

no questions

---

> ### Author Response · Authors · 2024-11-20
> **Response to Reviewer znzb**
>
> Dear Reviewer,
>
> Thank you for time to provide such detailed and thoughtful feedback on our work. We would like to address your concerns one by one.
>
> ### Comment 1: "The emulator-driven fine-tuning provides only binary correctness feedback..."
>
>
> > The emulator-driven fine-tuning provides only binary correctness feedback on the predicted code, which might not be sufficient for identifying and correcting specific errors in the generated code. Can it be in natural language?
>
> You are absolutely right in pointing out that our emulator-driven fine-tuning technique currently provides binary correctness feedback. This emulator's feedback is used to determine the likelihood of a sample being included in the training process for the next epoch.
>
> Regarding your suggestion about providing feedback in natural language, our emulator is indeed capable of generating natural language feedback, similar to the error messages typically seen in Python. These execution messages could be leveraged to calculate the probability of a sample being used, depending on the nature of the error. For example, if the error is minor, we could increase the likelihood of including the sample, whereas for more significant errors, the probability could be reduced.
>
> This is a valuable suggestion, and we recognize its potential. However, given that our current focus is on datasets and benchmarking model performance, we have left the exploration of natural language feedback for error correction as a topic for future work. We appreciate your insightful feedback, and it will certainly be considered in the direction of our future research.
>
>
> ---
>
> ### Comment 2: "There's a risk that models could overfit to the synthetic data..."
>
> > There's a risk that models could overfit to the synthetic data used in training, which might not perfectly mimic the complexity and variability of real-world programming challenges. I think this is the most serious disadvantage
>
> We appreciate the concern regarding the potential for models to overfit to synthetic data.
>
> To understand if our model was overfitting to synthetic data, we analyzed its performance across different epochs (Figure 15 in the Appendix). We observed that the model's performance on the synthetic dataset (SIM-EVAL) continued to improve after epoch 6, while its performance on the real-world dataset (REAL) plateaued. If we continue to fine-tune the model on the synthetic training dataset for more epochs, it might overfit to the synthetic data. Consequently, we limited the fine-tuning to 8 epochs to mitigate overfitting. On the other hand, we acknowledge that generating synthetic data that perfectly mimics the complexity and variability of real-world programming challenges is inherently difficult, and we will continue to address this issue in our future work.
>
> ---
>
> Thank you for your valuable feedback. We hope our response has addressed your concerns and positively influenced your evaluation. We appreciate your time and effort to improve this work!

---

> ### Author Response · Authors · 2024-11-25
> **A gentle reminder**
>
> We hope that our responses have addressed the reviewer's concerns and questions. Given that the discussion period is ending soon, we would greatly appreciate the reviewer's input. If you have more questions, we would be glad to provide further responses. Thank you for your feedback!

---

> > ### Comment · Reviewer_znzb · 2024-11-27
> > **Reply to authors**
> >
> > Thanks for your feedback, I decide to increase my score!

---

> > > ### Author Response · Authors · 2024-11-27
> > >
> > > Thank you for your valuable feedback and for raising your score. We have made improvements to the manuscript since then. If you have any remaining concerns that might prevent you from increasing your score further, please let us know. We would be happy to address them!

---

### Official Review · Reviewer_miWe · 2024-11-08

**Soundness:** 3
**Presentation:** 4
**Contribution:** 2
**Rating:** 5
**Confidence:** 4

**Summary:**

This paper
1. curates a small (85 tasks) program synthesis benchmark based on real-world tasks in the XLogoOnline visual programming environment, which requires a combination of different skills;
2. shows that current SotA models like GPT-4V and Llama3-70B struggle to solve such tasks;
3. develops a data synthesis pipeline with curriculum learning based on task difficulty for the model to significantly boost the model performance on the benchmark.

**Strengths:**

1. This paper identifies a new task that current SotA models fail on.

**Weaknesses:**

1. This paper provides few new insights about identifying models' limitations: This paper shows that existing models fail in visual programming in XLogoOnline. But this task is similar to some tasks in works like MMMU, which also combine different skills like vision and reasoning and have already shown that current models struggle in visual reasoning tasks
2. This paper also lacks novelty about improving models' performance:
    1. The data synthesis method is based on sampling and filtration, which has been widely adopted in works like STaR, RaFT and Rest-EM.
    2. The curriculum learning design adaptive to task difficulty has been explored by works like DART-Math.

**Questions:**

1. To the best of my knowledge, OpenAI has never disclosed the parameter sizes of their models since GPT-3.5. Where is the statement of "GPT3.5 model with 175B parameters (version gpt-3.5-turbo-0125)" (line 335-336) from?
2. Which version of Llama3 models are used? 3 or 3.1?

---

> ### Author Response · Authors · 2024-11-20
> **Response to Reviewer miWe (1/2)**
>
> Dear Reviewer,
>
> Thank you for your time to provide such detailed and thoughtful feedback on our work. We are grateful for your constructive comments and suggestions. We would like to address each of your comments in detail in the following.
>
> ---
>
> ### Comment 1: "This paper provides few new insights about identifying models' limitations..."
>
> > This paper provides few new insights about identifying models' limitations: This paper shows that existing models fail in visual programming in XLogoOnline. But this task is similar to some tasks in works like MMMU, which also combine different skills like vision and reasoning and have already shown that current models struggle in visual reasoning tasks
>
> We appreciate this observation and would like to emphasize the key differences between our benchmark and MMMU:
>
> 1. **Focus on Different Domains:**
>    - MMMU addresses broader challenges in _multimodal understanding and reasoning_, whereas our benchmark focuses on _program synthesis_ for _visual programming tasks_. This fundamental difference in focus also leads to differences in task types.
>    - For example, MMMU tasks primarily involve multiple-choice and open-ended questions with unambiguous correct answers. In contrast, our benchmark tasks require models to generate executable code snippets for grid navigation. These solution codes are not required to be unique but must adhere to specific coding and grid constraints to achieve the desired goals. As far as we know, elements such as writing code, navigating grids, and satisfying constraints are absent in MMMU.
>
> 2. **Emphasis on different Skill Sets:**
>    - While both benchmarks test visual understanding and reasoning, our benchmark introduces additional challenges requiring programming skills, planning, and spatial reasoning. These aspects are not explored in MMMU.
>
> Moreover, our analysis (see Section 5.3) reveals that *spatial reasoning remains a significant weakness in large models*, despite their strong performance in other domains. We believe that our benchmark provides valuable insights into the limitations of large models, particularly in this underexplored area.
>
>
> ---
>
> ### Comment 2: "This paper also lacks novelty about improving models' performance..."
>
>
> > This paper also lacks novelty about improving models' performance:
> > - The data synthesis method is based on sampling and filtration, which has been widely adopted in works like STaR, RaFT and Rest-EM.
>
>
> We would like to clarify the differences between our approach and existing methods such as STaR [1], RaFT [2], and Rest-EM [3]:
>
> 1. **Answer-First Synthesis Approach:**
>    - Unlike the question-first approaches used in STaR [1] and Rest-EM [3], where questions are generated first and answers synthesized later, we adopt an **answer-first approach**.
>    - In our method, we start by synthesizing the answer (i.e., code) and then derive the question (i.e., task) to ensure it aligns with the synthesized answer. This approach is particularly effective in our domain, as existing LLMs are not well-trained on domain-specific data. Using a question-first approach would result in poor performance, as demonstrated in Table 6, where Llama3-70B achieves only 2.35% accuracy in generating correct domain-specific answers.
>
> 2. **Symbolic Execution and Constraint Solving:**
>    - Our synthesis technique does not rely on LLMs for generating data, as seen in STaR [1], RAFT [2], and Rest-EM [3]. Instead, we leverage **symbolic execution and constraint solving**.
>    - Specifically, we begin with a reference (task, code) pair. By mutating the reference code, we generate variations and then synthesize tasks solvable by the mutated code. This ensures the generated data is both correct and tailored to the benchmark, a critical distinction from existing works.

---

> > ### Author Response · Authors · 2024-11-20
> > **Response to Reviewer miWe (2/2)**
> >
> > ### Comment 3: "The curriculum learning design adaptive to task difficulty has been explored by works like DART-Math."
> >
> >
> >   > The curriculum learning design adaptive to task difficulty has been explored by works like DART-Math.
> >
> > Thank you for bringing this up. However, we would like to clarify that there are key differences between our approach and DART-Math:
> >
> > 1. **Different Objectives:**
> >    - DART-Math focuses on _dataset generation_, while our curriculum learning design is specifically aimed at _fine-tuning models_. Our approach adaptively adjusts the distribution of training data for each epoch of model training, making it complementary to DART-Math rather than overlapping.
> >
> > 2. **Different Motivations:**
> >    - DART-Math is motivated by the observation that hard questions are underrepresented in datasets because their answers are difficult to obtain. Its goal is to generate more hard questions.
> >    - In contrast, our motivation stems from the observation that hard tasks are challenging for models to learn early in training. Therefore, we begin by training models on easier tasks and progressively increase task difficulty as the models improve. This process is guided by an emulator that provides feedback.
> >
> > ---
> >
> > ### Comment 4: "Clarifications on GPT-3.5 and Llama3 versions..."
> >
> >
> > > 1. To the best of my knowledge, OpenAI has never disclosed the parameter sizes of their models since GPT-3.5. Where is the statement of "GPT3.5 model with 175B parameters (version gpt-3.5-turbo-0125)" (line 335-336) from?
> > > 2. Which version of Llama3 models are used? 3 or 3.1?
> >
> >
> > Thank you for pointing this out. We have made the following clarifications and corrections:
> >
> > 1. **Parameter Sizes of GPT-3.5:**
> >    - We agree with your point. The statement about GPT-3.5-turbo-0125 having 175B parameters was derived from other research papers, such as [4]. However, upon further review, we could not find any official OpenAI document confirming this information. As such, we have removed this statement from the updated version of the paper.
> >
> > 2. **Llama3 Version:**
> >    - We used **Llama3** in our experiments, not Llama3.1.
> >
> > We appreciate your attention to these details and have revised the paper accordingly about the parameter sizes of GPT-3.5. Thank you once again for your valuable feedback, which has helped us improve our paper. Please let us know if you have additional questions or concerns, and we will be happy to address them.
> >
> >
> > **References**
> >
> > [1]: Eric Zelikman, Yuhuai Wu, Jesse Mu, Noah D. Goodman:
> > STaR: Bootstrapping Reasoning With Reasoning. NeurIPS 2022.
> >
> > [2]: Tianjun Zhang, Shishir G. Patil, Naman Jain, Sheng Shen, Matei Zaharia, Ion Stoica, Joseph E. Gonzalez:
> > RAFT: Adapting Language Model to Domain Specific RAG. CoRR abs/2403.10131.
> >
> > [3]: 	Avi Singh, et al. Beyond Human Data: Scaling Self-Training for Problem-Solving with Language Models. Trans. Mach. Learn. Res. 2024.
> >
> > [4]: Zhijing Jin, et al. CLADDER: Assessing Causal Reasoning in Language Models. NeurIPS 2023.

---

> > ### Comment · Reviewer_miWe · 2024-11-20
> >
> > Thanks for your reply!
> >
> > ### Lack of new information
> >
> > I admit there exist some difference between this benchmark and other visual reasoning benchmarks. However, the key problem is that it brings **little new information**.
> >
> > In a practical view, a benchmark is meaningful if it measures some abilities needed in real-world tasks. However, as Reviewer KKKK mentioned in his/her Questions, the benchmark is not so "real-world" but a synthetic toy.
> >
> > In an analytical view, a benchmark is meaningful if it provides insights about the internal mechanics of the system. However, the paper lacks such analysis and only shows that models are weak on this task and can be improved by training on a lot of similar data, which is quite trivial and has been clearly shown by previous works like MMMU and LLaVA.
> >
> > ### The training method is probably new but highly limited
> >
> > Given the differences you mentioned, your training method might differ from previous methods, but this is based on the special task environment and, thus, highly limited, making little contribution.

---

> > > ### Author Response · Authors · 2024-11-20
> > >
> > > We thank the reviewer this the prompt reply and further feedback. We would like to address the points raised one by one.
> > >
> > > ---
> > >
> > > > However, the key problem is that it brings **little new information**.
> > > > In a practical view, a benchmark is meaningful if it measures some abilities needed in real-world tasks. However, as Reviewer KKKK mentioned in his/her Questions, the benchmark is not so "real-world" but a synthetic toy.
> > >
> > >
> > > While we understand the reviewer’s concern, we believe that our benchmark does provide new information, especially in the programming education domain. To clarify, our benchmark tasks are collected from the XLogoOnline platform, which is widely used in programming education for young students (e.g., K-2 students). Each year, **tens of thousands of students** uses this platform [1], making our tasks reflective of real-world educational scenarios.
> > >
> > > We believe the unique aspect of our benchmark is that, **despite these tasks being trivial and easy for human experts, they are surprisingly challenging for even the most advanced models.**
> > >
> > > ---
> > >
> > > > In an analytical view, a benchmark is meaningful if it provides insights about the internal mechanics of the system. However, the paper lacks such analysis and only shows that models are weak on this task and can be improved by training on a lot of similar data, which is quite trivial and has been clearly shown by previous works like MMMU and LLaVA.
> > >
> > > While we understand the reviewer’s concern, we would like to clarify that we don't just show that models are weak on our tasks. We had provided many different of analysis in our paper:
> > > - **First, we had provided very detailed failure analysis (See Section 5.3 and Appendix D.4).** We found that spatial reasoning is the most significant weakness for why state-of-the-art models fail, which is not provided in MMMU or Llava.
> > > - **Second, we had provided a comparative analysis of models’ capabilities (See Section 5.4, paragraph 2).**
> > > - **Third, we had provided insights into whether fine-tuned models are capable of learning transferable skills (see Section 5.4, paragraph 3).**
> > >
> > >
> > > ---
> > >
> > > > Given the differences you mentioned, your training method might differ from previous methods, but this is based on the special task environment and, thus, highly limited, making little contribution.
> > >
> > > Our curriculum-learning based method requires only the emulator to provide feedback, which is used to adjust the training data distribution. Therefore, **in domains where an "emulator" or a "critic" exists, our method can also be applied.** For example, when fine-tuning a model using a collection of Python tasks and codes, we can use the Python interpreter as the emulator to provide feedback and dynamically adjust the training data distribution.
> > >
> > > ---
> > >
> > > We thank the reviewer for their feedback and are happy to address any further questions.
> > >
> > >
> > > **References:**
> > >
> > > [1]: Juraj Hromkovič, Giovanni Serafini & Jacqueline Staub. XLogoOnline: A Single-Page, Browser-Based Programming Environment for Schools Aiming at Reducing Cognitive Load on Pupils.

---

> ### Author Response · Authors · 2024-11-25
> **A gentle reminder**
>
> We hope that our responses have addressed the reviewer's concerns and questions. Given that the discussion period is ending soon, we would greatly appreciate the reviewer's input. If you have more questions, we would be glad to provide further responses. Thank you for your feedback!

---

> > ### Comment · Reviewer_miWe · 2024-11-25
> >
> > Thanks for your reply! I agree this paper does find a countercase for VLMs, but all the information is not quite surprising to me. I would raise the score but I still take it as below the acceptance line.

---

> > > ### Author Response · Authors · 2024-11-26
> > >
> > > Thank you so much for your reply and raising the score. We appreciate your feedback and perspective on the work!

---

### Author Response · Authors · 2024-11-21

Dear Reviewers and AC,

We would like to express our sincere gratitude to all the reviewers and the AC for their feedback and constructive suggestions on our manuscript. During the rebuttal period, we have focused on addressing these valuable feedback, incorporating additional experiments, and refining our manuscript. We hope that our current revised manuscript can address all reviewers' concerns.

As the reviewers highlighted, we believe our paper introduces a novel benchmark for the visual programming domain (**Reviewer znzb, Reviewer nRED, Reviewer KKKK**), which is easy for humans yet challenging for current SOTA models (**Reviewer KKKK**).

We appreciate the reviewers' feedback that our manuscript is well-written and organized (**Reviewer miWe, Reviewer 81e9, Reviewer nRED, Reviewer KKKK**), provides sufficient details on datasets and experiments (**Reviewer nRED**), and offers analysis that provides new insights (**Reviewer nRED, Reviewer KKKK**).

We also thank the reviewers for raising concerns regarding the challenges posed by our benchmark (**Reviewer miWe, Reviewer 81e9**), the evaluation of additional multimodal models (**Reviewer 81e9, Reviewer nRED**), the potential overfitting on synthetic data (**Reviewer znzb**), the clarification of the data generation process and emulator (**Reviewer nRED**), prompt engineering (**Reviewer KKKK**), and the impact of fine-tuning on other benchmarks (**Reviewer KKKK**).

In response, these concerns have been addressed in the rebuttal. Additionally, we have also carefully revised and enhanced our manuscript with the following key updates:

- **[All Reviewers]** We moved the Appendix from the Supplementary Material (zip file) to the end of the references in the main paper to make it easier for reviewers to access.
- **[Reviewer miWe]** We have removed the mention of GPT-3.5 parameters from the paper (Section 5.1).
- **[Reviewer 81e9, Reviewer nRED]** We added the evaluation of 7 additional state-of-the-art multimodal models (Figure 6).
- **[Reviewer nRED]** We added details of the emulator (Appendix C).
- **[Reviewer KKKK]** We added examples of tasks from the synthetic SIM dataset (Figure 12).
- **[Reviewer KKKK]** We added additional experiments on the impact of fine-tuning on other benchmarks (Appendix D.4).

For easier review, these updates are temporarily highlighted in **orange** in our manuscript.

We hope our responses and revisions address all the reviewers' concerns, and we are eager to engage in further discussions regarding these updates.

Thank you,
The Authors

---

### Author Response · Authors · 2024-12-02
**General reminder**

Dear Reviewers,

We thank all the reviewers for their comments and feedback. As the discussion period ends in less than 24 hours, we hope our responses below have addressed the concerns raised in your reviews. If there are any remaining issues or points that need clarification, we would be happy to address them within the remaining time.

Thank you,
The Authors

---

### Meta-Review · Area_Chair_qW8Q · 2024-12-24

**Metareview:**

The paper introduces a new benchmark to evaluate the performance of LLM on program synthesis tasks within the XLogoOnline visual environment. The primary concerns raised include:

1. Limited technical novelty

2. Insufficient experimental evaluation

 While the paper has notable merit, it is not recommended for acceptance. The authors are advised to incorporate the reviewers' feedback when revising the paper for submission to other venues.

**Additional Comments On Reviewer Discussion:**

NA

---

### Decision · Program_Chairs · 2025-01-22

Reject